# Elastic force restricts growth of the murine utricle

**Ksenia Gnedeva[1,2]\*[†], Adrian Jacobo[1†], Joshua D Salvi[1], Aleksandra A Petelski[1], A J Hudspeth[1]**

[1]Howard Hughes Medical Institute and Laboratory of Sensory Neuroscience, The Rockefeller University, New York, United States; [2]Eli and Edythe Broad Center for Regenerative Medicine and Stem Cell Research, Keck School of Medicine of the University of Southern California, Los Angeles, United States

**Abstract** Dysfunctions of hearing and balance are often irreversible in mammals owing to the inability of cells in the inner ear to proliferate and replace lost sensory receptors. To determine the molecular basis of this deficiency we have investigated the dynamics of growth and cellular proliferation in a murine vestibular organ, the utricle. Based on this analysis, we have created a theoretical model that captures the key features of the organ's morphogenesis. Our experimental data and model demonstrate that an elastic force opposes growth of the utricular sensory epithelium during development, confines cellular proliferation to the organ's periphery, and eventually arrests its growth. We find that an increase in cellular density and the subsequent degradation of the transcriptional cofactor Yap underlie this process. A reduction in mechanical constraints results in accumulation and nuclear translocation of Yap, which triggers proliferation and restores the utricle's growth; interfering with Yap's activity reverses this effect.

**\*For correspondence:** gnedeva@usc.edu

[†]These authors contributed equally to this work

**Competing interests:** The authors declare that no competing interests exist.

## Introduction

The sensory organs of the inner ear arise from patches of Sox2-positive cells specified in the prosensory domain of the otic vesicle (*Kiernan et al., 2005*; *Hartman et al., 2010*). Despite differences in function, the organs for hearing and balance have similar structures and are populated by the same type of mechanosensory receptor, the hair cell. Hair cells are intercalated with supporting cells that are necessary for the proper sensory functions (*Haddon et al., 1999*). Additionally, when hair cells are lost in nonmammalian species, supporting cells can proliferate and transdifferentiate into new sensory receptors, allowing for recovery of hearing and balance (*Corwin and Cotanche, 1988*; *Ryals and Rubel, 1988*; *Harris et al., 2003*; *Taylor and Forge, 2005*). Although supporting cells in the vestibular sensory organs of adult mammals retain a limited ability to regenerate hair cells through direct conversion (*Ruben, 1967*; *Forge et al., 1993*; *Rubel et al., 1995*; *Kawamoto et al., 2009*; *Lin et al., 2011*; *Golub et al., 2012*), they lose the ability to reenter the cell cycle after neonatal stages (*Golub et al., 2012*; *Burns et al., 2012a*; *Wang et al., 2015*; but see *Warchol et al., 1993*). This deficiency may be a primary reason why hearing and balance function fail to recover in mammals after hair cell damage.

In order to understand what blocks the proliferation of supporting cells during regeneration, it is important to uncover the mechanism that arrests cellular proliferation in the developing sensory epithelia of the inner ear. The signaling pathways that control the proliferation of supporting cells during development have been studied extensively. For example, proliferation is blocked by downregulation of Wnt signaling in the developing murine inner ear (*Jacques et al., 2012*; *Chai et al., 2012*). ErbB signaling is also involved, for EGF and heregulin enhance supporting-cell proliferation in vitro and inhibition of the EGFR pathway arrests mitotic activity in sensory epithelia

(**Zheng et al., 1999**; **Hume et al., 2003**; **Doetzlhofer et al., 2004**; **White et al., 2012**). We have recently demonstrated that SoxC transcription factors play an important role and that reactivation of their expression in the adult utricle elicits supporting-cell proliferation and the production of hair cells (**Gnedeva and Hudspeth, 2015**). Maturational changes in cell-cell and cell-matrix adhesion have also been shown to correlate with the ability of supporting cells to proliferate and transdifferentiate into hair cells in the utricle (**Davies et al., 2007**; **Burns et al., 2008**; **Collado et al., 2011**). Finally, although it is not clear what triggers their expression, the upregulation of retinoblastoma protein and of cyclin-dependent kinase inhibitors such as p27Kip1 enforces a nonproliferative state in postmitotic supporting cells (**Chen and Segil, 1999**; **Löwenheim et al., 1999**; **Yu et al., 2010**). However, neither conditional ablation of p27Kip1 nor forced re-expression of cyclin D in adult sensory epithelia is sufficient to relieve the block on supporting-cell proliferation (**Laine et al., 2010**; **Loponen et al., 2011**). These observations suggest the existence of parallel repressive mechanisms.

In this work, we used a vestibular sensory organ—the utricle—as a model system to study the cell-cycle exit in the sensory epithelia of the inner ear. By combining theoretical and experimental approaches to recreate the known features of utricular organogenesis, we elucidated a previously unrecognized mechanism underlying the arrest of the organ's growth.

## Results

### Dynamics of utricular macula development

To assess the dynamics of the sensory epithelial growth during development in the murine utricle, we monitored the organ's development from embryonic day 15.5 (E15.5) through postnatal day 14 (P14). We observed a rapid expansion of the utricle's sensory area—the macula—as determined by Sox2 labeling, between E15.5 and E18.5 (**Figure 1A,D**). After E19.5 the rate of macular growth declined until the organ reached its final size at P2. As was shown previously (**Burns et al., 2012b**), the rate of areal growth $\dot{A}(t)$ as a function of time fits the von Bertalanffy growth equation $\dot{A}(t) = r[A_\infty - A(t)]$ in which $r$ is a rate constant, $A_\infty$ is the final area, and $A(t)$ is the area at time $t$. The solution to this equation is $A(t) = A_\infty e^{r(t-t_0)}$, in which $t_0$ is an integration constant (**Figure 1D**). This equality indicates that the growth rate decreases continuously from the outset to zero when the tissue has reached its target size $A_\infty$. Although this relation implies that the rate is negatively regulated by an increase in the organ's size, it offers no information about the underlying mechanism. We therefore examined several models for self-regulation of the utricle's sensory epithelia growth, focusing on the two that proved most effective.

To reduce the mathematical complexity of the system we introduced a two-dimensional model that captures the main qualitative features of the developing utricle (**Figure 1—figure supplement 1**). In this representation, the sensory epithelium of the utricle is surrounded by an elastic boundary that comprises all the nonsensory tissues encompassing the macula. We assume that the boundary is a linear elastic material that can be described by its Young's modulus, or elastic modulus. This material property is defined as the ratio of the stress, or force per unit area, to the strain, the ratio of the deformation over the initial length, and indicates how much the material resists deformations imposed by an external force. We then used a lattice-based approach to create a stochastic computational simulation of macular growth (**Swat et al., 2012**).

In our first representation—the elasticity-limited model—we employed a value of the elastic modulus $E$ such that the tissue is restricted to its experimentally observed size solely by the force exerted by the elastic boundary (**Figure 1B,E**; **Video 1**; **Figure 1—source data 1**; **Source code 1**; **Table 1**). In this model, the growth of the utricle ceases when the pressure generated by stretching of the elastic boundary equals that created by the growth and division of supporting cells. In this case the final size of the utricle is inversely related to the square of the elastic modulus $E$ of the elastic boundary (**Figure 1F**; **Figure 1—source data 2**).

In our second representation—the morphogen-limited model—we retained a weaker elastic boundary and assumed that the supporting cells of the utricle secrete a growth-inhibiting molecule (**Figure 1C,E**; **Video 2**; **Figure 1—source data 3**; **Source code 2**; **Table 1**). We assume that the morphogen diffuses slowly from its cellular source. Once the concentration $[M]$ of the growth inhibitor reaches a certain threshold value $[M_{TH}]$ at the location of a given cell, this cell stops proliferating. The delay between supporting-cell formation and production of the inhibitor produces a gradient

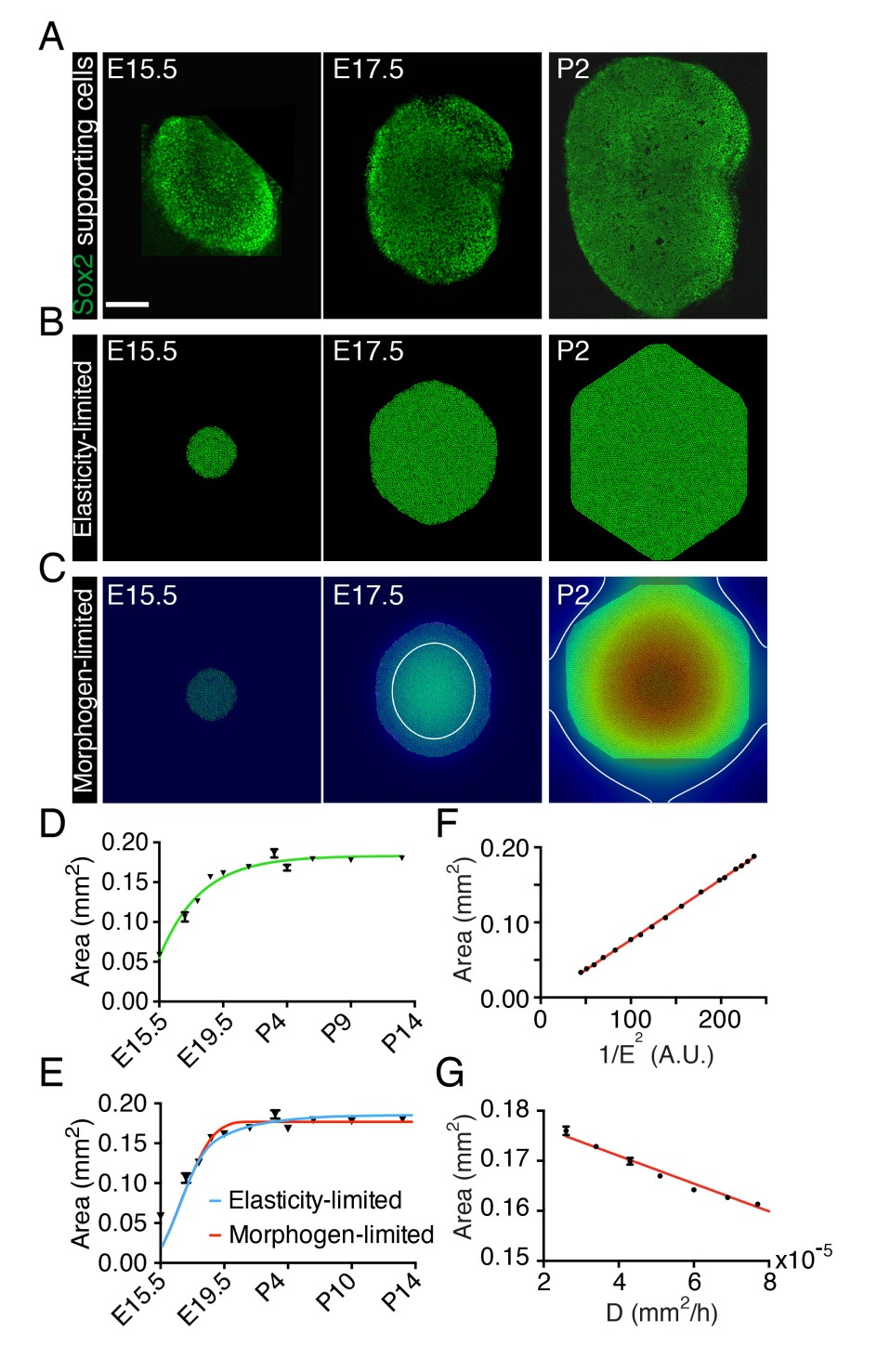

**Figure 1.** Size control in utricular development. (A) Immunolabeling of supporting cells with anti-Sox2 (green) demonstrates the size of the utricular sensory epithelia of mice at E15.5, E17.5, and P2. The scale bar represents 100 μm. (B) Simulations of the elasticity-limited model portray sensory epithelia at the same developmental stages (*Source code 1*; *Table 1*). (C) In simulations of the morphogen-limited model, the concentration of the diffusible morphogen [M] is shown on a spectral scale from blue as zero to red as the maximum, 0.2 pM. The white curve bounds the region in which a concentration exceeding the threshold value $[M]_{TH}$ = 0.04 pM inhibits cellular proliferation (*Source code 2*; *Table 1*). (D) The measured areas of sensory epithelia at different developmental stages (black triangles; means ± SEMs; *Figure 1—source data 1*) are fit by the von Bertalanffy equation (green

*Figure 1 continued on next page*

*Figure 1 continued*

curve; $A_\infty = 1.8 \cdot 10^{-7} \pm 5.8 \cdot 10^{-9}$ m$^2$, $r = 3.6 \cdot 10^{-6} \pm 5.8 \cdot 10^{-7}$ s$^{-1}$, $t_0 = -1.32 \cdot 10^{-5} \pm 3.7 \cdot 10^{-6}$ s$^{-1}$, $R^2 = 0.94$). (E) The same data are portrayed with fits of the elasticity-limited model (blue curve, $R^2 = 0.74$) and morphogen-limited model (red curve, $R^2 = 0.62$). Each result represents the average of five realizations; the SEM is typically around 1% (*Figure 1—source datas 3,5*). (F) Final areas of utricular sensory epithelia from simulations of the elasticity-limited model (means ± SEMs, $N = 3$) for different values of the Young's modulus of the elastic boundary $E$. The areas are proportional to $1/E^2$ (black dots) and fit the linear relation $Y = 0.000804 \cdot X - 0.00361$ (red line, $R^2 = 0.99$; *Figure 1—source data 2*). (G) Final areas of utricular sensory epithelia from simulations of the morphogen-limited model (means ± SEMs, $N = 3$) are linearly related to the diffusion coefficient $D$ (black dots) by the relation $Y = -278 \cdot X - 0.182$ (red line, $R^2 = 0.98$; *Figure 1—source data 4*).

The following source data and figure supplement are available for figure 1:

**Source data 1.** The areas of utricular sensory epithelia measured at different developmental stages.
**Source data 2.** Final areas of utricular sensory epithelia from simulations of the elasticity-limited model for different values of the Young's modulus of the elastic boundary $E$.
**Source data 3.** Fits of the elasticity-limited model to the areas of utricular sensory epithelia measured at different developmental stages.
**Source data 4.** Final areas of utricular sensory epithelia from simulations of the morphogen-limited model for different values of the diffusion coefficient $D$.
**Source data 5.** Fits of the morphogen-limited model to the areas of utricular sensory epithelia measured at different developmental stages.
**Figure supplement 1.** Geometry of the utricle and simplification to a mathematical model.

with the highest concentration at the center of the utricular macula and a lower concentration at the periphery. Cells at the center therefore stop proliferating first and a front of cell-cycle arrest (white line in *Figure 1C*) expands until it reaches the edge of the sensory epithelium and fully stops its growth. In this case, the final size of the sensory epithelium largely depends on the secretion rate and diffusion coefficient of the morphogen and in its absence the tissue would grow to a much greater size than is observed experimentally (*Figure 1G*; *Figure 1—source data 4*).

Although the elasticity-limited model represents a better fit to the data, both models can explain the developmental growth curve of the utricle (*Figure 1E*; *Figure 1—source datas 3,5*). We therefore experimentally tested additional predictions of the two models to evaluate their validity.

## The effect of elastic force on the utricle's growth in vitro

To distinguish between the proposed models, we developed an ex vivo culture system that allows manipulation of the stiffness of the extracellular matrix surrounding the utricle while monitoring the organ's growth. For these experiments we used Atoh-nGFP transgenic mice (*Lumpkin et al., 2003*) in which the hair cells are labeled by GFP expression, to visualize the sensory epithelium. We dissected part of the

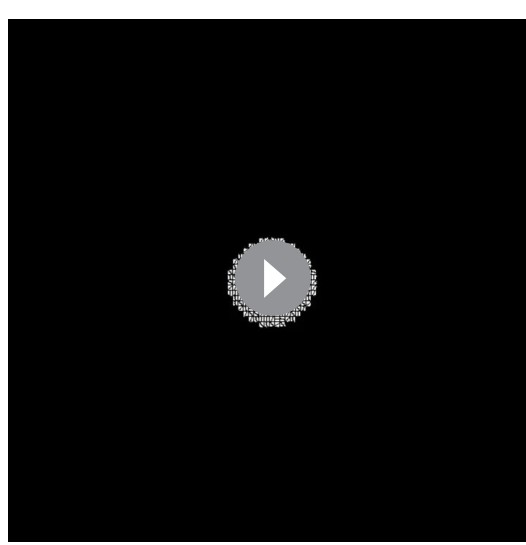

**Video 1.** A simulation of the elasticity-limited model corresponding to *Figure 1B* spans developmental stages E15.5 to P14 at 1 hr intervals.

Table 1. Parameter values for simulations of cellular proliferation in the utricle.

| Description | Parameter name | Value |
| --- | --- | --- |
| System size | $N$ | 600 × 600 (hexagonal pixels) |
| Amplitude of cell-membrane fluctuations | $T_m$ | 5 (dimensionless) |
| Cellular target volume | $V_T$ | 40 pixels |
| Cellular elastic modulus | $\lambda_{VOL}$ | 9 (dimensionless) |
| Cellular target surface area | $S_T$ | 25 pixels |
| Cellular membrane elastic modulus | $\lambda_{SURF}$ | 1.5 (dimensionless) |
| Elastic matrix's Young's modulus<br>Elasticity-limited model<br>Morphogen-limited model | $E$ | <br>0.069 (dimensionless)<br>0.055 (dimensionless) |
| Cell-cell interaction energy | $J_C$ | 20 (dimensionless) |
| Morphogen diffusion coefficient | $D$ | $1.19 \cdot 10^{-14}$ m$^2 \cdot$s$^{-1}$ |
| Morphogen secretion rate | $k_M$ | 0.23 s$^{-1}$ |
| Morphogen concentration threshold | $[M]_{TH}$ | 0.1 pM |
| Time after division for cells to become quiescent | $T_{div}$ | 2000 (Monte Carlo steps) |
| Mean refractory time | $T_{ref}$ | 200 (Monte Carlo steps) |
| Refractory time standard deviation | $\sigma_{ref}$ | 50 (Monte Carlo steps) |
| Monte Carlo step time | $dt$ | 210 s |
| Model's spatial scale | $dA$ | $8.6 \cdot 10^{-13}$ m$^2$ |

vestibular apparatus—the utricle, two ampullae, and portions of the semicircular canals—at E17.5 and maintained it as a free-floating culture for 3 hr. Healing of the cuts introduced during dissection yielded a closed organotypic structure that we term a utricular bubble (*Figure 2A*). Such a culture reduces the leakage of any secreted molecules from the system, allowing us to test the predictions of the morphogen-limited model.

To change the force opposing the sensory epithelia growth, we embedded the utricular bubble in a collagen gel of calibrated stiffness (*Figure 2B*). In our mathematical model this manipulation equates to changing the stiffness of the elastic boundary surrounding the macula. Because the developing inner ear is normally encapsulated by cartilage, we increased the elastic modulus of the gels by cross-linking the collagen fibers with addition of chondrocytes (*Figure 2—figure supplement 1*). A gel with an elastic modulus of 640 ± 3 Pa, corresponding to the stiffness of supporting cells in the inner ear (*Sugawara et al., 2004*; *Zetes et al., 2012*), was used to approximate the stiffness of the elastic boundary that the utricle experiences in vivo. At the opposite extreme, a 40 ± 6 Pa gel with no chondrocytes created significantly less force on the growing utricle.

Although both our models predicted that reducing the stiffness of the elastic boundary would allow the sensory epithelium to expand to

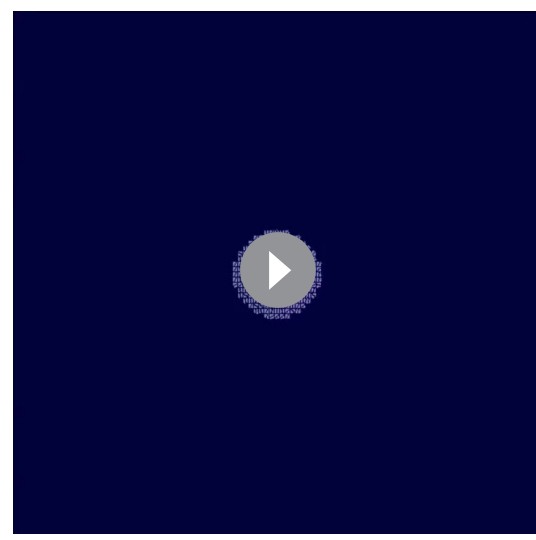

**Video 2.** A simulation of the morphogen-limited model corresponding to *Figure 1C* spans developmental stages E15.5 to P14 at 1 hr intervals. The concentration of the diffusible morphogen is shown on a spectral scale from blue as zero to red as the maximum, 0.20 pM. The white curve bounds the region in which a concentration exceeding the threshold value = 0.04 pM inhibits cellular proliferation.

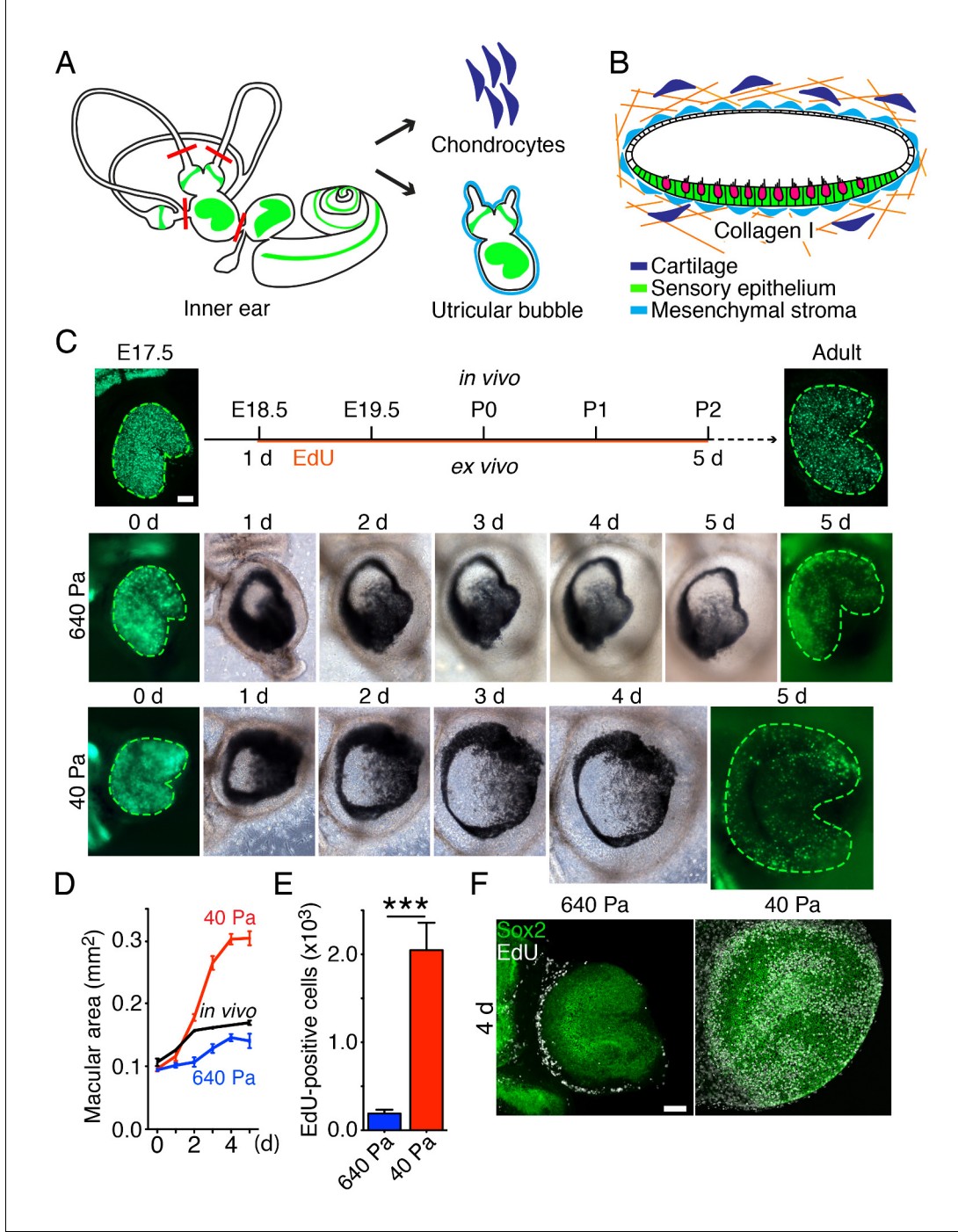

**Figure 2.** Ex vivo utricular culture system. (**A**) A schematic drawing portrays the sensory epithelia (green) of the six receptor organs of the murine inner ear. The red lines delineate the cuts introduced during the dissection of a utricular bubble. The mesenchymal stroma that normally surrounds the organ is shown in light blue; the chondrocytes that surround the inner ear are shown in dark blue. (**B**) In a diagram of a section through a utricular bubble, hair cells are shown in magenta, supporting cells in green, non-sensory cells in white, and mesenchyme in light blue. The bubble is embedded in a gel of collagen I (orange) containing chondrocytes (dark blue). (**C**) A sequence of surface views portrays the development of two utricular bubbles from E17.5 Atoh-nGFP mice over a period of 5 d. The sensory epithelia are marked in the first and final images by GFP expression (green; delineated by green dashed line). In a gel of 640 Pa stiffness (middle panels), the utricular sensory epithelium reaches the normal adult size (in vivo; upper panels) by 5 d, corresponding to developmental stage P2. In a collagen gel of 40 Pa stiffness (lower panels) the area of the sensory epithelium almost doubles by 5 d. The scale bar represents 50

*Figure 2 continued*

μm. (D) Areal measurements, represented as means ± SEMs, demonstrate that the utricular sensory epithelium in a 640 Pa gel develops at a rate similar to that of the epithelium in vivo. By 5 d in culture the area of the sensory epithelium is similar to that of an adult utricle (p<0.05, *N* = 5 for both; *Figure 2—source data 1*). In a 40 Pa gel, the area of the sensory epithelium expands significantly faster and reaches 180% of the control value by 5 d (p<0.0001, *N* = 4 for 40 Pa; *N* = 5 for in vivo measurements; *Figure 2—source data 1*). (E) After 4 d in culture, quantification of EdU-positive cells in 0.1 $mm^2$ of the utricular sensory epithelium, represented as means ± SEMs, demonstrates a tenfold increase in the number of EdU-positive cells in a 40 Pa gel as compared with a 640 Pa gel. This difference is significant (p<0.001, *N* = 3 for both; *Figure 2—source data 2*) (F) A surface view of utricular bubbles after 4 d in culture demonstrates that the expansion of the macula in a 40 Pa gel stems from the re-entry of Sox2-positive supporting cells (green) into the mitotic cycle throughout the sensory epithelium, as demonstrated by the incorporation of EdU (white). In the identical preparation of the utricular bubble cultured in a 640 Pa gel, supporting-cell proliferation is limited to the organ's periphery. The scale bar represents 50 μm.

The following source data and figure supplements are available for figure 2:

**Source data 1.** The area of the sensory epithelium of the utricles cultured in 40 Pa and 640 Pa gels for 1–5 d.

**Source data 2.** The number of EdU-positive cells in 0.1 $mm^2$ of the utricular sensory epithelium in 40 Pa and 640 Pa gels after 4 d in culture.

**Figure supplement 1.** Characterization of the ex vivo utricular culture system.

**Figure supplement 2.** Chondrocyte-secreted factors do not affect supporting-cell proliferation.

---

a larger size, the expected spatial distribution of proliferating cells differed between the models. Because the concentration of growth-inhibiting morphogen should not be affected by external force, the morphogen-limited model predicted that the cells at the center of the utricle would not proliferate, regardless of a decrease in the gel's stiffness. On the contrary, if cell growth is suppressed only by elastic force, as proposed by the elasticity-limited model, supporting-cell proliferation should occur even near the center of the utricle.

We observed that in a 640 Pa gel, the E17.5 utricle grew at a rate similar to that found during normal development, reaching the size of P2 utricle by 5 d in culture (*Figure 2C,D*; *Figure 2—source data 1*). In contrast, over the same period a substantial expansion of a utricle's macular area occurred in a 40 Pa collagen gel, where the organ reached 180% of its normal adult size (*Figure 2C, D*; *Figure 2—source data 1*). Quantification of supporting-cell proliferation by EdU incorporation over 4 d in culture revealed a significant tenfold increase in 40 Pa gels as compared to 640 Pa gels (*Figure 2E*; *Figure 2—source data 2*). In agreement with the elasticity-limited model, we observed that whereas the proliferation was limited to the utricle's periphery in 640 Pa gels, supporting cells throughout the sensory epithelium reentered the cell cycle in 40 Pa gels (*Figure 2F*).

To ensure that the presence of chondrocytes did not inhibit supporting-cell proliferation, we cultured utricles in 40 Pa gels with chondrocytes situated locally at one edge of the organ. When after 24 hr we assessed the number of EdU-positive cells, we found no effect of the cartilage on the proliferation of supporting cells (*Figure 2—figure supplement 2*).

## Yap signaling during development and regeneration of the murine utricle

Simulations of the elasticity-limited model suggest that elastic force arrests the growth of the utricle, but leaves the organ in an unstable state; if cells retain their capacity to proliferate, random fluctuations could evoke additional growth. Because this behavior is not observed in vivo, elastic force likely triggers a secondary mechanism that stops proliferation when the organ reaches its target size. Hippo signaling, which is known to arrest cellular proliferation in response to mechanical force (*Aragona et al., 2013*; *Dupont et al., 2011*; *Robinson and Moberg, 2011*), might provide such a mechanism. Activation of Hippo signaling results in phosphorylation and degradation of the transcriptional cofactor Yap, restricting it from the nucleus and curtailing cellular proliferation. To

investigate the potential role of this pathway in the utricle's growth control, we assessed the expression of Yap, Hippo signaling kinases, and downstream target genes during the utricle's development.

The growth of the murine utricle and supporting-cell proliferation decline dramatically within the first two days of postnatal life (*Figure 1A,D*; *Burns et al., 2012b*; *Gnedeva and Hudspeth, 2015*). During the following week, the organ matures and the supporting cells within it lose the ability to reenter the cell cycle after damage (*Kawamoto et al., 2009*; *Golub et al., 2012*; *Burns et al., 2012a*; *Wang et al., 2015*). Our RNA-sequencing data (*Gnedeva and Hudspeth, 2015*; GSE72293) indicate that the expression of genes encoding the key components of the pathway—*Mst1* and *Mst2*, *Lats1* and *Lats2*, *Yap*, and *Taz*—was similar in actively growing utricles at E17.5 and growth-arrested mature utricles at P9 (*Table 2*). However, consonant with the activation of Hippo signaling, the level of Yap protein decreased significantly from E17.5 to P14 (*Figure 3A*). The expression of genes encoding the downstream targets of nuclear Yap, such as *Ankrd1*, *Ctgf*, and *Cyr61* (*Aragona et al., 2013*), was accordingly downregulated significantly by P2 (*Figure 3B*; *Figure 3—source data 1*). In addition, the expression of genes encoding known inhibitors of nuclear Yap translocation, such as E-cadherin, α-catenin, and gelsolin (*Aragona et al., 2013*; *Robinson and Moberg, 2011*), was upregulated postnatally as the organ matured and lost its capacity for proliferative regeneration (*Figure 3C*; *Figure 3—source data 2*).

To investigate the relationship between the subnuclear localization of Yap and supporting-cell proliferation in the developing utricle, we analyzed the inner ears of E17.5 Atoh-nGFP mice. A single EdU injection 4 hr prior to analysis was used to label proliferating supporting cells. We demonstrated cytoplasmic labeling of Yap in postmitotic supporting cells near the center of the utricular sensory epithelium (*Figure 3D,E*; *Figure 3—source data 3*). The proliferating cells at the organ's periphery showed significantly stronger Yap labeling, indicating some nuclear translocation of the protein. Yap was not present in hair cells.

The supporting cells in the utricle of a neonatal mouse retains a limited capacity to reenter the cell cycle (*Burns et al., 2012a*; *Wang et al., 2015*). To assess the potential role of Yap during this process, we examined the protein's expression in P4 utricles after injury in vitro. Utricles were dissected and allowed to attach to the bottom of a Petri dish coated with Cell-Tack adhesive. Subsequently, a linear cut was made with a 30-gauge needle along one border of the sensory epithelium, and each utricle was allowed to recover for 48 hr in culture medium containing EdU (*Figure 4A*). As expected (*Davies et al., 2007*; *Meyers and Corwin, 2007*), Sox2-positive supporting cells reentered

**Table 2.** Expression ratios for relevant genes in utricular sensory epithelia in late embryonic development (E17.5) and after maturation (P9).

| Gene | RPKM E17.5 | RPKM P9 | Expression ratio E17.5/P9 | *p*-value | Function |
|------|-----------|---------|---------------------------|-----------|----------|
| *Dach1* | 12.0 | 5.7 | 2.0 | <0.05 | Cytoskeletal interactors |
| *Nf2* | 25.0 | 27.4 | 0.91 | >0.05 | |
| *Wwc1* | 76.2 | 72.5 | 1.0 | >0.05 | |
| *Sav1* | 17.8 | 27.2 | 0.65 | >0.05 | Kinases |
| *Stk3* | 18.3 | 19.7 | 0.92 | >0.05 | |
| *Stk4* | 13.6 | 11.4 | 1.1 | >0.05 | |
| *Lats1* | 26.5 | 26.5 | 1.0 | >0.05 | |
| *Lats2* | 5.2 | 3.6 | 1.4 | >0.05 | |
| *Mob1a* | 27.1 | 20.9 | 1.2 | >0.05 | |
| *Mob1b* | 13.9 | 17.5 | 0.79 | >0.05 | |
| *Taz* | 21.5 | 29.8 | 0.72 | >0.05 | Transcriptional coactivators |
| *Yap1* | 53.9 | 63.5 | 0.84 | >0.05 | |

The complete data set is available through GEO database accession number GSE72293. RPKM, reads per kilobase of transcript per million mapped reads.

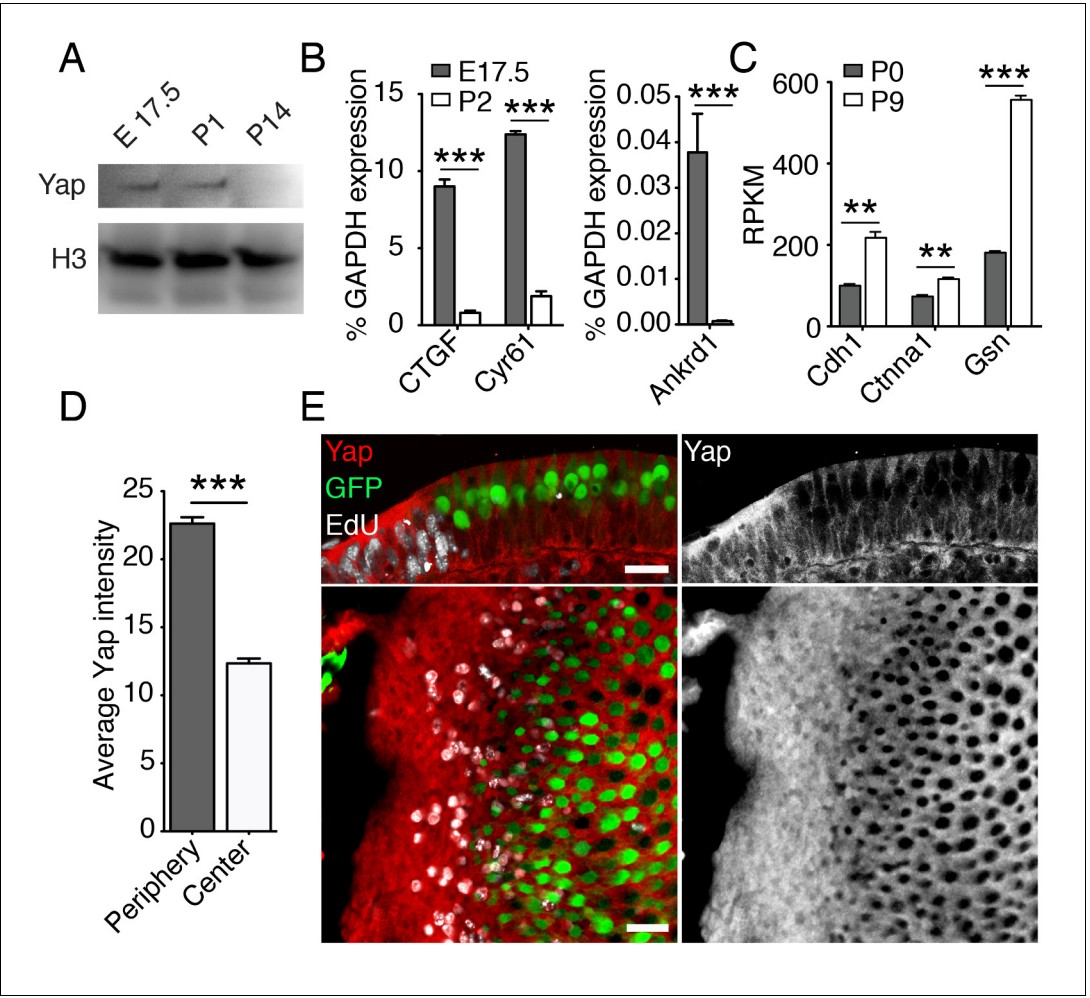

**Figure 3.** Yap signaling during utricular development. (**A**) Western blot analysis of the total protein extracted from normally developing murine utricles demonstrates a progressive decrease in the level of Yap protein. (**B**) qPCR analysis of normally developing murine utricles reveals significant decreases in the expression of the Yap target genes *Ctgf*, *Cyr61*, and *Ankrd1* between E17.5 and P2 (means ± SEMs; p<0.001 for each; $N = 6$ for E17.5; and $N = 9$ for P2; *Figure 3—source data 1*). (**C**) RNA sequencing from normally developing murine utricles reveals a progressive increase in the expression of genes encoding inhibitors of Yap tranlocation to the nucleus—*Cdh1*, *Ctnna1*, and *Gsn*—between P0 and P9 (means ± SEMs; p<0.005 for *Cdh1*, p<0.005 for *Ctnna1*, and p<0.0001 for *Gsn*; $N = 3$ for each; *Figure 3—source data 2*). (**D**) The mean fluorescence intensity of antibody labeling for Yap is significantly higher at the periphery than near the center of the utricle (value range 0–85; means ± SEMs; p<0.0001; $N = 4$ for each; *Figure 3—source data 3*) (**E**) In cross-sections (top panels) of the peripheral utricular sensory epithelium of E17.5 Atoh-nGFP mice 4 hr after EdU treatment, antibody labeling demonstrates Yap protein (red at left; white at right) in actively proliferating cells labeled with EdU (white at left). Hair cells (green) do not express Yap. In a different specimen this pattern of labeling is demonstrated in surface views (bottom panels). The scale bar represents 25 μm.

The following source data is available for figure 3:

**Source data 1.** qPCR analysis of the expression of the Yap target genes *Ctgf*, *Cyr61*, and *Ankrd1* at E17.5 and P2.

**Source data 2.** RNA sequencing of the expression of genes encoding inhibitors of Yap tranlocation to the nucleus—*Cdh1*, *Ctnna1*, and *Gsn*—at P0 and P9.

**Source data 3.** The mean fluorescence intensity of antibody labeling for Yap at the periphery and near the center of the E17.5 utricle.

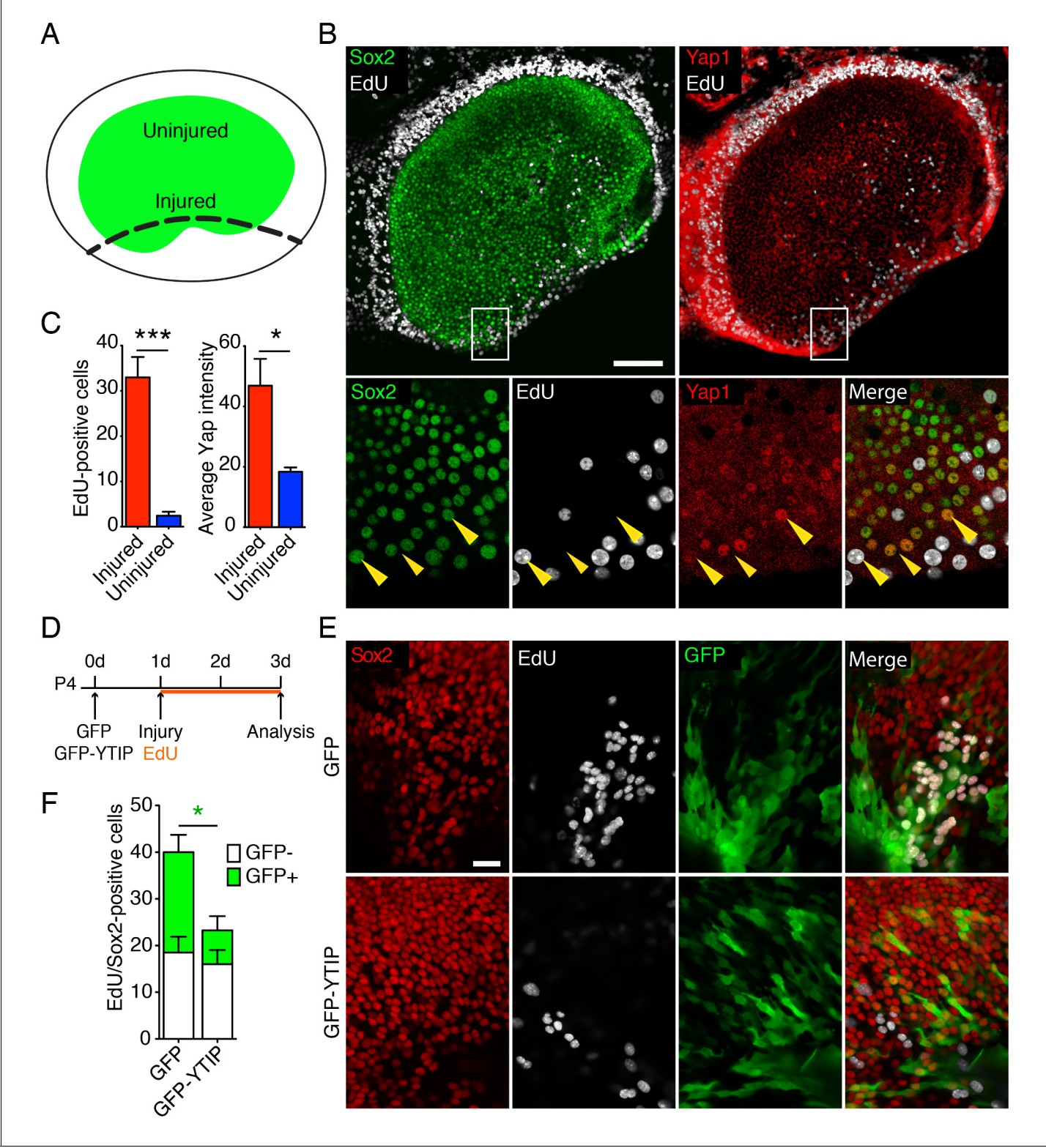

**Figure 4.** Yap distribution during utricular regeneration after injury in vitro. (**A**) A schematic drawing portrays the sensory epithelium (green) bounded by non-sensory epithelial cells (white). The black dashed line delineates the lesion. (**B**) In P4 utricles cultured for 48 hr in vitro, Sox2-positive supporting cells (green) at the injury site, as well as the transitional non-sensory epithelial cells that bound the sensory epithelium, express high levels of Yap protein (red). As shown by the incorporation of EdU (white), the same cells re-enter the cell cycle. Yap nuclear translocation can be seen in individual Sox2-positive supporting cells at the injury site (magnified 4X in lower panels; yellow arrowheads). The scale bar represents 100 μm. (**C**) The number of

*Figure 4 continued on next page*

*Figure 4 continued*

EdU-positive cells in the injured areas of utricular sensory epithelia rises significantly in comparison to noninjured areas of the same epithelia (cells counted over 0.01 mm$^2$; means ± SEMs; p<0.001, N = 4 for injured areas and N = 8 for noninjured areas; *Figure 4—source data 1*). The amount of proliferation correlates with the average intensity of Yap immunolabeling, which is significantly higher in injured areas (value range 0–85; means ± SEMs; p<0.05, N = 6 for injured areas and N = 5 for noninjured areas; *Figure 4—source data 2*). (D) A schematic representation of the lesioning experiment demonstrates the time at which the utricles are transfected with GFP and GFP-YTIP virus, the time of injury, and the duration of EdU labeling. (E) In P4 utricles cultured for 3 d in vitro, Sox2-positive supporting cells (red) at the injury site enter the cell cycle, as shown by EdU incorporation (white). Cells expressing GFP and GFP-YTIP are shown in green. The scale bar represents 25 μm. (F) Although the number of GFP-negative proliferating supporting cells remains unchanged between the conditions (white bars; cells counted over 0.01 mm$^2$; means ± SEMs; p>0.05, N = 4 for each; *Figure 4—source data 3*), the number of GFP-positive proliferating supporting cells is significantly reduced at the injury site of the utricles expressing GFP-YTIP as compared to cntrols expressing GFPs (green bars; cells counted over 0.01 mm$^2$; means ± SEMs; p<0.025, N = 4 for each; *Figure 4—source data 3*).

The following source data and figure supplement are available for figure 4:

**Source data 1.** The number of EdU-positive cells in the injured and noninjured areas of P4 utricular sensory epithelium after 24 hr in culture.

**Source data 2.** The mean fluorescence intensity of antibody labeling for Yap in the injured and noninjured areas of P4 utricular sensory epithelium after 24 hr in culture.

**Source data 3.** The number cells doubly positive for EdU and Sox2 in the injured areas of P4 utricular sensory epithelium infected with virus carrying GFP or GFP-YTIP.

**Figure supplement 1.** Yap controls supporting-cell proliferation in vitro.

the cell cycle at the site of injury (*Figure 4B,C*). Accordingly, as demonstrated by significant increase in the intensity of Yap antibody labeling (*Figure 4C*; *Figure 4—source data 1* and *2*), we observed the accumulation of Yap protein in the proliferating sensory epithelium, where Sox2-positive supporting cells translocated Yap to the nuclei (*Figure 4B*).

We next sought to determine whether the ability of supporting cells to reenter the cell cycle after injury depends on nuclear Yap signaling. In most contexts, Yap must bind Tead transcription factors to stimulate cellular proliferation (*Vassilev et al., 2001*). We therefore overexpressed GFP fused to a Yap-Tead interfering peptide, YTIP (*von Gise et al., 2012*) in ex vivo cultures. By using the Ad-Easy cloning system (*Chartier et al., 1996*; *He et al., 1998*), we developed a serotype five adenoviral overexpression vector carrying GFP-YTIP. We tested the transfection efficiency and toxicity of the virus on E16.5 utricle explants in vitro and found no difference with respect to a control virus expressing only GFP (*Figure 4—figure supplement 1*). The number of proliferating cells in the sensory epithelia of the utricles infected with GFP-YTIP was greatly reduced, reaching only 7% that in GFP controls (*Figure 4—figure supplement 1*).

We tested the effect of GFP-YTIP overexpression in the injury assay. After the utricles had been dissected and affixed to the bottom of a Petri dish, the cultures were infected with control GFP virus or with GFP-YTIP virus and left for 24 hr to allow the accumulation of GFP and GFP-YTIP proteins (*Figure 4D*). On the following day, the injury assay was performed and each utricle was allowed to recover for 48 hr in culture medium containing EdU. As determined by Sox2 and EdU labeling, the number of GFP-negative proliferating supporting cells at the injury site in utricles infected with GFP-YTIP virus was not significantly different from that in control GFP cultures. However, the number of GFP-positive cells within the proliferating supporting-cell population was reduced significantly in GFP-YTIP cultures as compared to GFP controls (*Figure 4E,F*; *Figure 4—source data 3*).

## Yap signaling downstream of elastic force

The pattern of Yap expression in the utricle during development and after injury suggested that the protein regulates supporting-cell proliferation. To test whether nuclear Yap translocation is controlled by elastic force we assessed protein expression in collagen gels of varying stiffness. The elasticity-limited model predicts that reducing the stiffness of the elastic boundary would allow cells in the center of the organ to expand in volume and therefore to reenter the cell cycle. A significant increase in the area of the utricular macula was observed in 40 Pa gels as compared to 640 Pa gels after 2–3 d in culture (*Figure 2D*). In agreement with the model's predictions, we observed a

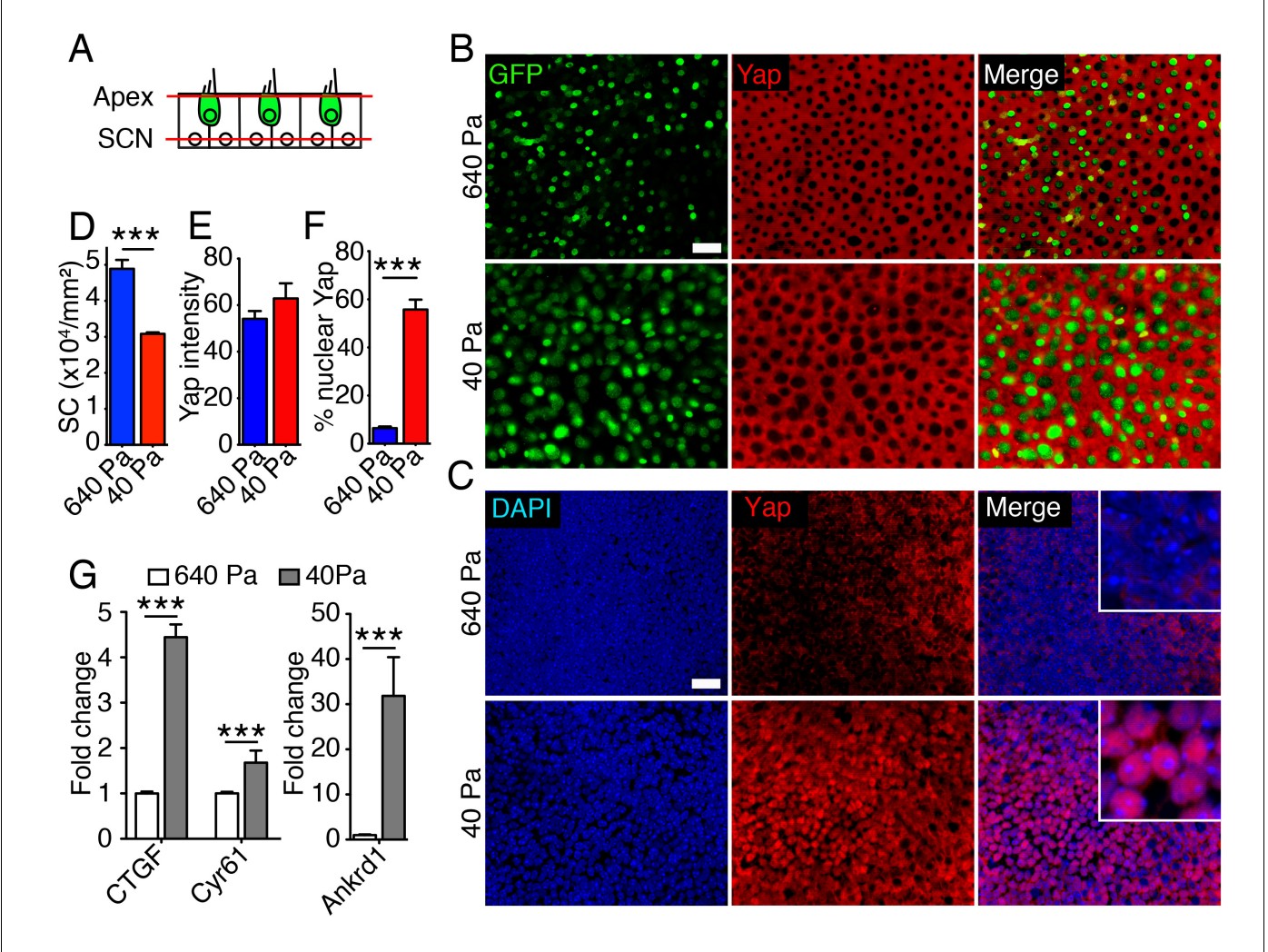

**Figure 5.** Subcellular localization of Yap in organotypic bubble cultures. (A) In a schematic representation of a cross-section through an E17.5 Atoh-nGFP utricle, red lines represent optical sections through the apices of hair and supporting cells (Apex; shown in panel B) and through supporting cell nuclei (SCN; shown in panel C). Hair cells are marked in green. (B) In surface views of utricles maintained for 3 d as organotypic bubble cultures in collagen gels of Young's modulus 640 Pa (top panels) or 40 Pa (bottom panels), cytoplasmic Yap protein occurs in supporting cells but is nor observed in hair cells (green). The scale bar represents 25 µm. (C) In the same organotypic bubble cultures, Yap protein is excluded from the nuclei of supporting cells in a 640 Pa collagen gel (top panels). In contrast, Yap is translocated into the nuclei of supporting cells in a 40 Pa gel (bottom panels). The insets in the merge panels are magnified fourfold. The scale bar represents 25 µm. (D) Quantification of the density of Sox2-positive cells demonstrates a significant decrease in 40 Pa gels as compared to 640 Pa gels (means ± SEMs; p<0.001; N = 3 for each; *Figure 5—source data 1*). (E) The mean fluorescence intensity of antibody labeling for Yap is unchanged in utricles cultured for 3 d in 40 Pa gels as compared to 640 Pa gels (value range 0–85; means ± SEMs; p>0.05; N = 5 for 640 Pa and N = 6 for 40 Pa; *Figure 5—source data 2*). (F) The percentage of supporting cells showing Yap nuclear translocation is significantly higher in 40 Pa gels than in 640 Pa gels (means ± SEMs; p<0.0001; N = 4 for each; *Figure 5—source data 3*). (G) qPCR analysis of utricle bubble cultures maintained for 4 d reveals significant increase in the expression of the Yap target genes *Ctgf*, *Cyr61*, and *Ankrd1* in 40 Pa gels as compared to 640 Pa gels (means ± SEMs; p<0.001 for each; N = 10 for each; *Figure 5—source data 4*).

The following source data and figure supplement are available for figure 5:

**Source data 1.** Quantification of the density of Sox2-positive cells in utricular bubble cultures maintained in 40 Pa and 640 Pa gels for 3 d.

**Source data 2.** The mean fluorescence intensity of antibody labeling for Yap in the supporting-cell cytoplasm in utricular bubble cultures maintained in 40 Pa and 640 Pa gels for 3 d.

**Source data 3.** The percentage of supporting cells showing Yap nuclear translocation in utricular bubble cultures maintained in 40 Pa and 640 Pa gels for 3 d.

*Figure 5 continued on next page*

*Figure 5 continued*

**Source data 4.** qPCR analysis of the expression of the Yap target genes *Ctgf*, *Cyr61*, and *Ankrd1* in utricular bubble cultures maintained in 40 Pa and 640 Pa gels for 3 d.

**Figure supplement 1.** Subcellular Yap localization in organotypic bubble cultures.

decrease in cellular density in 40 Pa gels as compared to 640 Pa gels (*Figure 5D*; *Figure 5—source data 1*). As shown by immunolabeling, Yap was localized to the cytoplasm in the supporting cells of utricular bubbles maintained for 3 d in both 640 Pa and 40 Pa gels (*Figure 5B,E*; *Figure 5—source data 2*). However, consistent with a role of Yap in the regulating the utricular growth, the protein translocated into the nuclei in over 50% of supporting cells only in 40 Pa gels (*Figure 5C,F*; *Figure 5—source data 3*). Whereas cell proliferation was limited to the organ's periphery in 640 Pa gels by 4 d in culture, the supporting cells throurgout the utricular macula reentered the cell cycle in 40 Pa gels (*Figure 2F*; *Figure 5—source data 3*). Yap labeling at the same time revealed that the pattern of the protein's nuclear translocation in both conditions was consistent with the pattern of supporting-cell proliferation (*Figure 5—figure supplement 1*). The expression of genes encoding the downstream targets of nuclear Yap, *Ankrd1*, *Ctgf*, and *Cyr61*, was also upregulated significantly in 40 Pa gels as compared to 640 Pa gels (*Figure 5G*; *Figure 5—source data 4*).

If the extensive growth and cellular proliferation observed in 40 Pa collagen gels are triggered by the nuclear translocation of Yap, perturbation of the protein's nuclear function should abrogate the effect of low stiffness. We therefore tested the effect of GFP-YTIP on the utricle's ability to grow in a low-stiffness gel. Bubble cultures were established in a 40 Pa collagen gel and left overnight to allow the attachment of mesenchymal stroma cells (*Figure 6A*). On the following day either GFP-YTIP or control GFP virus was injected into each utricular bubble and the cultures were left for an additional 24 hr to allow the accumulation of GFP or GFP-YTIP protein. EdU was then added to the medium and the utricular bubbles were cultured for an additional 48 hr. As determined by Sox2 labeling, the areas of the utricular maculae of GFP controls increased significantly as compared to GFP-YTIP cultures (*Figure 6B,C*; *Figure 6—source data 1*). In accord with our previous results, the areas of the sensory epithelia in GFP virus-infected bubble cultures were similar to those observed in uninfected cultures after 4–5 d (*Figure 2D*), whereas GFP-YTIP cultures stalled at a size typical of control cultures at 2–3 d. The increase in macular area was reflected by a significant increase in the number of EdU-positive supporting cells (*Figure 6B,D*; *Figure 6—source data 2*). The supporting-cell density in GFP-YTIP virus-infected utricles was significantly higher than that in GFP virus-infected control cultures (*Figure 6E*; *Figure 6—source data 3*). These results strongly suggest that reduction in elastic force triggers supporting-cell proliferation through nuclear Yap signaling.

## The effect of elastic force on the pattern of supporting-cell proliferation

To determine whether the elasticity-limited model can explain the rate and pattern of supporting-cell proliferation in vivo, we analyzed supporting-cell numbers during utricular development. The macular area of the organ expanded dramatically between E15.5 and E17.5 (*Figure 1*); the number of supporting cells concurrently quadrupled (*Figure 7A*; *Figure 7—source data 1*). Over the same period, we observed a significant increase in supporting-cell density (*Figure 7B*). The rate of growth decreased thereafter and cell numbers plateaued by P2, when supporting-cell proliferation decreases dramatically (*Burns et al., 2012b*; *Gnedeva and Hudspeth, 2015*). However, we did not detect a significant increase in cellular density from E18.5 to P2 (*Figure 7B*; *Figure 7—source data 2*).

To visualize the pattern of supporting-cell proliferation we performed an EdU pulse-chase experiment. Pregnant mice were injected at E17.5 and the inner ears of their progeny were analyzed 12 hr later. Consistent with previous reports (*Burns et al., 2012b*; *Gnedeva and Hudspeth, 2015*), recently divided supporting cells were localized at the periphery of the utricle's sensory epithelium, where small patches of EdU-positive cells were observed (*Figure 7C*). To quantify this phenomenon, we measured the average density of EdU-positive cells and the average outline of the macular

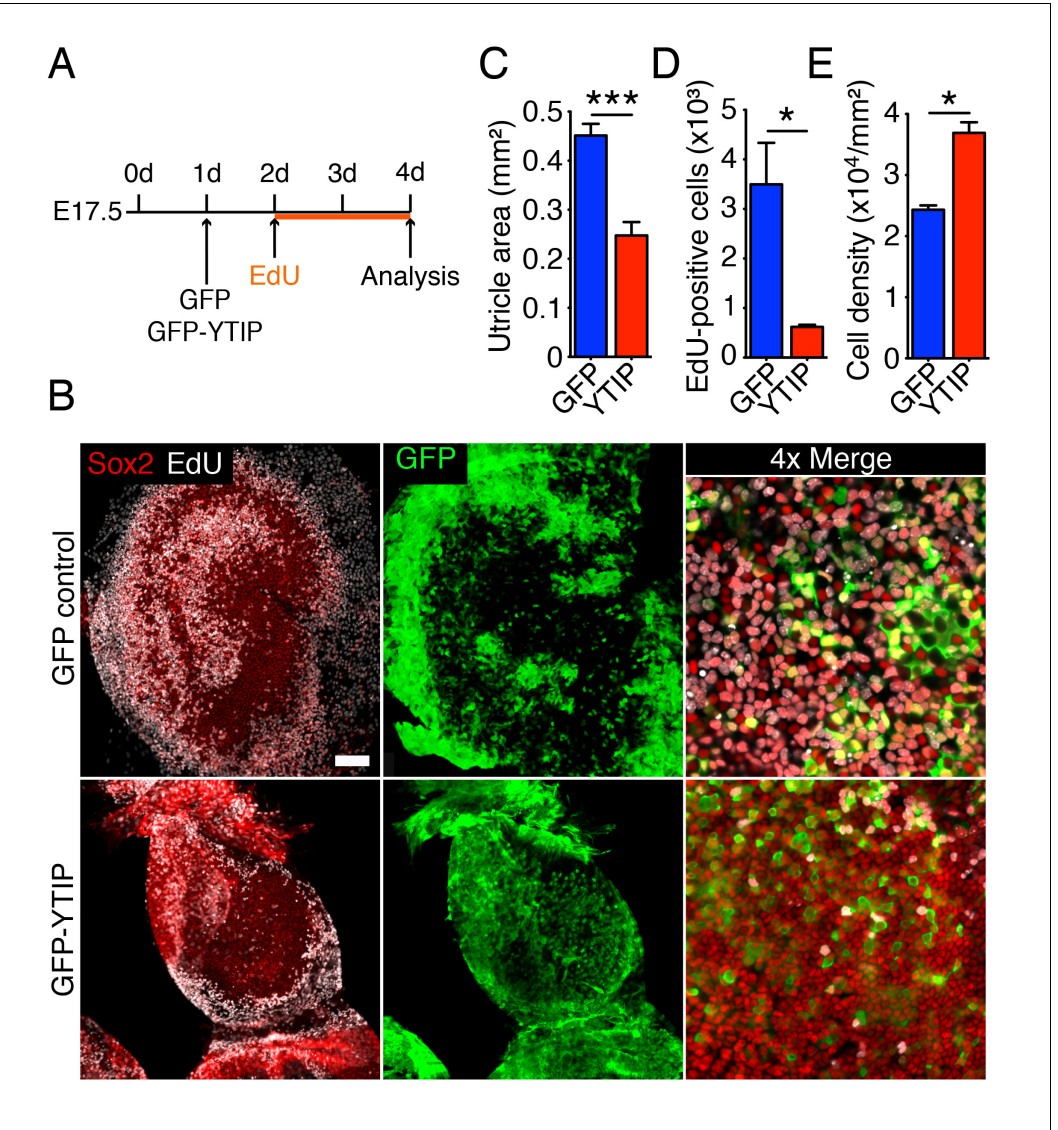

**Figure 6.** Yap controls utricular growth in organotypic bubble cultures. (**A**) A schematic representation of an experiment on an organotypic bubble culture shows the time at which the utricles are transfected with virus expressing GFP or GFP-YTIP and the duration of EdU labeling. (**B**) Sox2 antiserum (red) labels the sensory epithelia of whole-mount preparations of utricular bubbles infected with virus expressing GFP or GFP-YTIP. As demonstrated by the incorporation of EdU (white), supporting cells in E17.5 utricles in GFP virus-infected cultures, but not in GFP-YTIP virus-infected cultures, re-enter the cell cycle. GFP, which demonstrates infected cells, is shown in green. The scale bar represents 100 µm. (**C**) The area of the sensory epithelia and (**D**) the number of EdU-positive cells in GFP virus-infected controls are increased significantly as compared to GFP-YTIP virus-infected utricles (means ± SEMs; area p<0.001; cell number p<0.05; *N* = 4 for GFP and *N* = 3 for GFP-YTIP; *Figure 6—source data 1* and *2*). (**E**) The extent of proliferation is inversely correlated with supporting-cell density, which is significantly decreased in GFP virus-infected controls as compared to GFP-YTIP virus-infected utricles (means ± SEMs; p<0.05; *N* = 5 for GFP and *N* = 6 for GFP-YTIP; *Figure 6—source data 3*).

The following source data is available for figure 6:

**Source data 1.** The area of the sensory epithelia in GFP and GFP-YTIP virus-infected utricular bubble cultures maintained in 40 Pa gels for 4 d.

**Source data 2.** The number of EdU-positive supporting cells in GFP and GFP-YTIP virus-infected utricular bubble cultures maintained in 40 Pa gels for 4 d.

*Figure 6 continued on next page*

*Figure 6 continued*

**Source data 3.** The supporting-cell density in GFP and GFP-YTIP virus-infected utricular bubble cultures maintained in 40 Pa gels for 4 d.

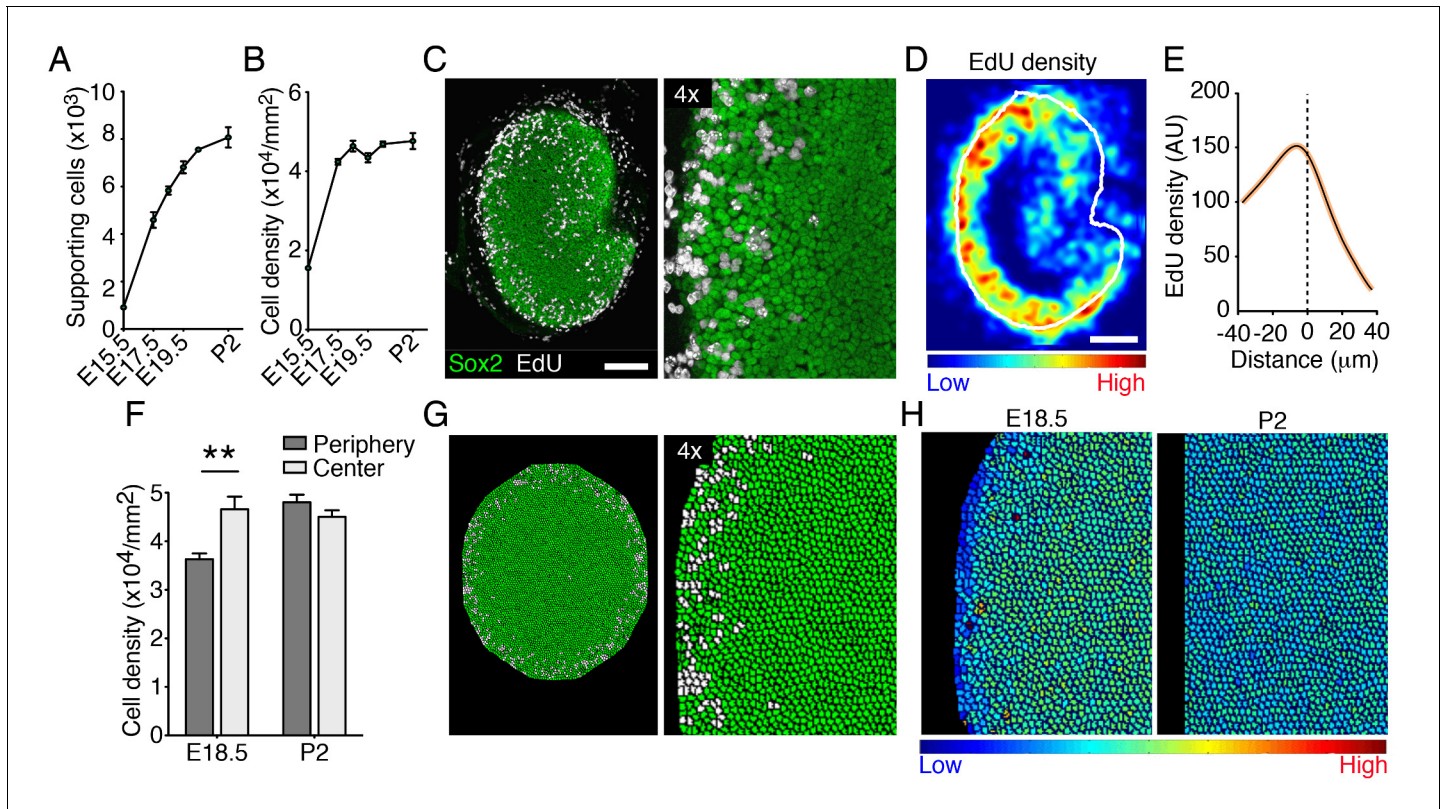

**Figure 7.** Cellular density and proliferation in the developing utricle. (A) The numbers of supporting cells in utricular sensory epithelia increases significantly from E15.5 to P2 (means ± SEMs; p<0.05, *N* = 5 for each; *Figure 7—source data 1*). (B) The supporting-cell density grew significantly from E15.5 to E17.5 (means ± SEMs; p<0.05, *N* = 5 for each; *Figure 7—source data 2*) and plateaus thereafter. (C) In the Sox2-positive utricular sensory macula (green) of an E18.5 embryo exposed to EdU 12 hr earlier, dividing supporting cells (white) can be seen at the organ's periphery. The scale bar represents 100 μm. (D) The intensity of EdU labeling and the outline of the utricular macula (white line) are averaged for five E18.5 utricles to demonstrate the consistent pattern of supporting-cell proliferation at the corresponding developmental stage. (E) The distribution of EdU intensity shown in (D) demonstrates a clear peak that lies immediately inward from the macular boundary (dashed line). The standard errors are less that 1% and are shown by the orange area around the curve (*Figure 7—source data 3*). (F) The supporting-cell density at E18.5 is significantly higher at the center of the utricular macula than that at the organ's periphery (means ± SEMs; p<0.01, *N* = 4 for each; *Figure 7—source data 4*). There is no significant difference in supporting-cell density between the periphery and the center of the macula at P2 (means ± SEMs; p>0.05; *N* = 4 for each; *Figure 7—source data 4*). (G) Simulations of EdU labeling in the elasticity-limited model at E18.5 resemble the distribution of proliferating supporting cells seen experimentally at the same developmental stage. (H) The same simulation demonstrates the gradient in internal cell pressure in the E18.5 utricle. Areas of low pressure, and therefore of low cellular density, occur at the periphery; high pressure is predicted at the macular center. The gradient in cellular pressure disappears at P2.

The following source data is available for figure 7:

**Source data 1.** The numbers of supporting cells in utricular sensory epithelia counted at different developmental stages.
**Source data 2.** The supporting-cell density in utricular sensory epithelia at different developmental stages.
**Source data 3.** The average distribution of EdU intensity along the outline of the utricular macula in E18.5 utricle.
**Source data 4.** The supporting-cell density at E18.5 and P2 at the center and the periphery of the utricular macula.

sensory epithelium in multiple utricles at E18.5 (*Figure 7D*). This analysis demonstrated a clear maximum in the EdU intensity near the border of the macula (*Figure 7D,E*; *Figure 7—source data 3*). Using our elasticity-limited model to simulate the EdU pulse-chase experiment, we found a remarkably similar pattern of supporting-cell proliferation (*Figure 7G*). We explored the computational simulations to explain this pattern.

In our model, cells act as linear springs: the smaller a cell is compared to its equilibrium volume, the larger its internal pressure. The elasticity-limited model predicts that the elastic force produced by macular expansion creates a gradient of cellular pressure, with higher values at the center of the sensory epithelium and lower ones at the periphery where proliferation occurs (*Figure 7H*). Because cells with higher internal pressure have smaller volumes, the model predicts that the pressure gradient creates a corresponding gradient in cellular density. Although the average density does not increase after E17.5 (*Figure 7B*), we found that the density at the center of the macula significantly exceeded that at the periphery at E18.5 (*Figure 7F*; *Figure 7—source data 4*). As predicted by the model and demonstrated by our experimental data, the gradient in cellular pressure and density disappears concurrently with the cessation of macular growth at P2 (*Figure 7F,H*). In accord with the role of Yap in supporting-cell proliferation, an increase in cellular density is known to activate Hippo signaling and the subsequent degradation of Yap (*Aragona et al., 2013*; *Dupont et al., 2011*; *Robinson and Moberg, 2011*).

Our computational simulations also permitted evaluation of the role of stem cells in the developing utricle. Numerical simulations of a model invoking the original population of stem cells as the only source of proliferation yielded a pattern of cell divisions inconsistent with the experimental observations (*Video 3*; *Source code 3*). In fact, LGR5-positive cells, proposed to be the putative stem-cell population in the utricle (*Wang et al., 2015*), are restricted to the striolar region by E15.5 and are therefore unlikely to contribute to growth at the organ's periphery.

## Discussion

In conjunction with published data (*Burns et al., 2012b*; *Gnedeva and Hudspeth, 2015*), our experimental results indicate that the sensory epithelium of the utricle grows primarily through the proliferation of supporting cells at its periphery. The rate of proliferation is self-regulated and decreases as the organ approaches its target size. To explore the mechanism underlying this process we implemented a combination of theoretical and experimental approaches.

Our main conclusion is that growth of the sensory epithelium is restricted by the elastic force produced by the surrounding nonsensory tissues. This force increases as the utricular macula expands in size, physically impeding cellular growth and proliferation. A similar mechanism has been shown to exist in the context of malignant growth, in which an external pressure can restrict the size of a tumor by physically opposing its growth and inhibiting cellular proliferation (*Helmlinger et al., 1997*; *Cheng et al., 2009*; *Montel et al., 2011*). In keeping with this idea, we demonstrate that reduction of the elastic force in utricular bubble cultures decreased the cellular density and allowed supporting cells to reenter the cell cycle and the sensory epithelium to nearly double in size. To our knowledge such an expansion has not been observed heretofore in any organ of the inner ear.

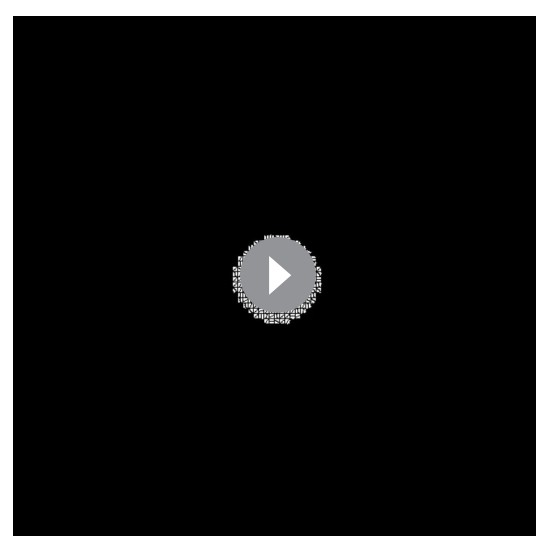

**Video 3.** A simulation of the theoretical model of macular development driven by a stem-cell population. In this variant of the elasticity-limited model, a patch of stem cells (blue) proliferates and differentiates into supporting cells (green). The model assumes that only stem cells divide (white). The model predicts that stem cells at the center of the utricular macula can proliferate only briefly before cellular density increases and precludes further divisions. Although some stem cells are advected towards the macular periphery, the pattern of proliferation is not consistent with that observed in vivo.

The elasticity-limited model not only explains the self-regulatory nature of macular growth but also suggests the involvement of the Hippo signaling pathway, which links mechanical force to cell-cycle arrest (*Dupont et al., 2011*; *Robinson and Moberg, 2011*; *Aragona et al., 2013*). By assessing the localization of Yap and supporting-cell density during normal development and in culture, we found that it is likely to control supporting-cell proliferation. Our simulations and data demonstrate that an elastic force, compressing the tissue, creates a cell-density gradient in the utricular macula, with a higher density at the center of the organ and a lower density at the periphery (*Figure 8*). High cellular density triggers the loss of nuclear Yap, causing supporting cells to exit the cell cycle. In support of this idea, removing the elastic constraints in organotypic utricular cultures decreased the supporting-cell density and resulted in nuclear Yap translocation and cell-cycle reentry throughout the sensory epithelium. Because all the sensory organs in the inner ear are subjected to mechanical constraints similar to those in the utricle, it is likely that Hippo signaling also controls cell-cycle exit in the developing organ of Corti. In fact, a mechanism of size control whereby mechanical force restricts the size of a tissue through activation of the Hippo signaling pathway has been conserved across a range of organs and species. Hippo signaling plays a major role in size control of the wing disk in *Drosophila* (*Hariharan, 2015*), determines the final size of the liver in mammals (*Dong et al., 2007*; *Camargo et al., 2007*), and regulates limb-bud regeneration in *Xenopus* (*Hayashi et al., 2014*).

Our experiments implicate a DNA-binding partner of nuclear Yap in the developing inner ear. When translocated to the nucleus, Yap can bind a variety of transcription factors to stimulate downstream gene expression (*Yagi et al., 1999*; *Vassilev et al., 2001*; *Ferrigno et al., 2002*). By specifically blocking the Yap-Tead interaction, we are able to arrest supporting-cell proliferation after injury and to reverse the effect of low stiffness on the expansion of utricular maculae in bubble cultures. As we demonstrated earlier (*Gnedeva and Hudspeth, 2015*), activation of SoxC transcription factors can restore proliferation even in utricles of young adult mice. SoxC proteins directly upregulate the expression of Tead2 (*Bhattaram et al., 2010*), which is highly expressed in the developing sensory epithelium and whose expression declines dramatically as supporting-cell proliferation ceases (*Gnedeva and Hudspeth, 2015*). In conjunction with the present results, this finding suggests that Hippo-initiated loss of the Yap-Tead transcription-factor complex is ultimately responsible for arresting the utricle's growth and limiting supporting-cell proliferation.

During the last decade there has been a renewed interest in the study of the role of mechanical forces and their interplay with molecular signaling during development (*Hernández-Hernández et al., 2014*; *Hamada, 2015*; *Pasakarnis et al., 2016*; *Dreher et al., 2016*). Our work constitutes a new example of such interactions and raises the possibility that the elusive mechanism that triggers supporting-cell proliferation after the loss of hair cells in nonmammalian species is

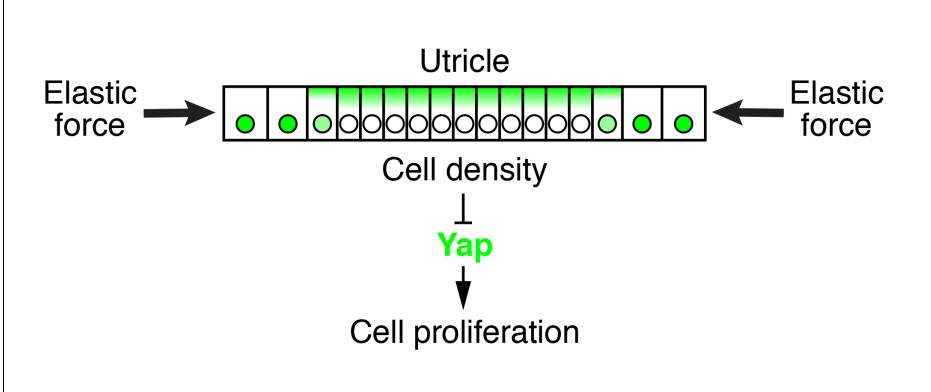

**Figure 8.** Elastic force restricts supporting-cell proliferation in the developing utricle. A schematic drawing portrays the sensory epithelium expanding in an elastic matrix. The force opposing utricle growth creates a gradient of cellular density, with higher values at the center of the organ and lower ones in the periphery. A high density triggers the cytoplasmic localization and degradation of Yap, whose consequent depletion from nuclei results in cell-cycle arrest.

mechanical in nature. Both the extrusion of dying hair cells from the sensory epithelium and the subsequent transdifferentiation might affect the mechanical force sensed by the residual supporting cells, causing them to re-enter the cell cycle. Although further investigation of the role of Hippo signaling in the inner ear is required, biochemical manipulation of this pathway might aid in the recovery of hearing and balance after the loss of hair cells.

## Materials and methods

### Animal care and strains

Experiments were conducted in accordance with the policies of The Rockefeller University's Institutional Animal Care and Use Committee and the Keck School of Medicine of the University of Southern California. Atoh1-nGFP mice were a kind gift from Dr. Jane Johnson. Swiss Webster mice with timed pregnancies were obtained from Charles River Laboratories.

### Dissection and culturing of utricles

Embryos were extracted from euthanized mice and placed into ice-cold Hank's balanced salt solution (HBSS, Life Technologies). Internal ears were dissected as described (*Gnedeva and Hudspeth, 2015*).

For organotypic cultures the ampullae of the anterior and horizontal semicircular canals and the nonsensory epithelium surrounding each utricle were left intact. The preparations were maintained for 3 hr at 37°C in complete growth medium comprising DMEM/F12 supplemented with 33 mM D-glucose, 19 mM NaHCO$_3$, 15 mM HEPES, 1 mM glutamine, 1 mM nicotinamide, 29 nM sodium selenite, 20 mg/L epidermal growth factor, 20 mg/L fibroblast growth factor, 10 mg/L insulin, and 5.5 mg/L transferrin (Sigma-Aldrich). Healing of the cut edges introduced during dissection allowed each preparation to reseal, creating an ellipsoidal structure termed a utricular bubble.

Collagen I was extracted from mouse-tail tendons (*Rajan et al., 2006*) and its concentration was adjusted to 2.0 mg/mL in 6 mM trichloroacetic acid (Sigma-Aldrich). 450 μL of collagen solution was mixed with 50 μL of 10X phosphate-buffered saline solution with phenol red pH indicator and neutralized by the addition of 11.9 mM NaOH and 1.3 mM NaHCO$_3$ to initiate polymerization. Utricular bubbles were placed into the collagen solution at room temperature and incubated for 20 min at 37°C to allow polymerization. The resultant cultures were then maintained at 37°C in complete growth medium equilibrated with 5% CO$_2$.

Chondrocytes were isolated from the cartilage surrounding the inner ear as described previously (*Gosset et al., 2008*).

### Immunohistochemistry

Utricles were dissected in ice-cold HBSS and fixed in 4% formaldehyde for 1 hr at room temperature. Whole inner ears were fixed for 18 hr at 4°C, treated with 0.88 M sucrose for 18 hr at 4°C, embedded in Tissue-Tek O.C.T. (Sakura), and frozen in liquid-nitrogen vapor. Wholemounted sensory epithelia or 10 μm frozen sections were then blocked with 3% normal donkey serum (Sigma-Aldrich) in 500 mM NaCl, 0.3% Triton X-100 (Sigma-Aldrich), and 20 mM tris(hydroxymethyl)aminomethane (Bio-Rad) at pH 7.5. The primary antisera—goat anti-Sox2 (Santa Cruz), rabbit anti-Myo7A (Proteus Bioscience), rabbit anti-GFP (Torrey Pines Biolabs), mouse anti-Yap (Santa Cruz), and rabbit anti-Yap (Cell Signaling)—were reconstituted in blocking solution and applied overnight at 4°C. For labeling with E-cadherin antibodies from clone DECMA-1 (Millipore) no Triton X-100 was used in the blocking solution.

Samples were washed with phosphate-buffered saline solution supplemented with 0.1% Tween 20 (Sigma-Aldrich), after which Alexa Fluor-labeled secondary antisera (Life Technologies) were applied in the same solution for 1 hr at room temperature.

Phalloidin conjugated to Alexa 633 was used to label filamentous actin and nuclei were stained with 3 μM DAPI.

## EdU labeling

EdU pulse-chase experiments were initiated by single intraperitoneal injections of 50 µg EdU (Life Technologies) per gram of body mass. Animals were sacrificed at the indicated times and the cells in the utricular sensory epithelia were analyzed by Click-iT EdU labeling (Life Technologies).

## Adenoviral gene transfer

The AdEasy Adenoviral Vector System (*Chartier et al., 1996*; *He et al., 1998*) was used to create adenoviral vectors containing the full-length coding sequence of green-fluorescent protein fused to the Yap-Tead interfering peptide (GFP-YTIP; Addgene plasmid 42238) under the control of a cyto-megalovirus promoter. Viral particles were amplified in HEK cells and purified by CsCl-gradient centrifugation followed by dialysis (Viral Vector Core Facility, Sanford-Burnham Medical Research Institute). Each utricle was dissected at E16.5-P4 and infected in 200 µL of culture medium with 10 µL of virus at a titer of $10^{10}$ PFU/mL. Alternatively, 1 µL of the virus was injected into the utricle in 3D organotypic cultures. Ad-GFP virus (Vector Biolabs) at the same titer was used as a control. One day later 3 mL of culture medium was supplemented with 10 µM EdU to label mitotic cells.

## RNA sequencing and qPCR analysis

The sample preparation and analysis for RNA sequencing have been described in detail (*Gnedeva and Hudspeth, 2015*). For qPCR, utricles at each developmental stage were isolated by microdissections and treated with 0.5% Dispase I (Sigma) for 15 min at 37˚C to isolate the sensory epithelia. For each sample, total RNA from 7 to 12 utricular maculae was isolated by a standard protocol (RNeasy Micro Kit, Qiagen) and used to create a cDNA library. The qPCR primers were designed with PrimerQuest (Integrated DNA Technologies). Relative gene-expression levels were obtained by normalization to the expression of Gapdh in each sample. qPCR analyses were performed on an Applied Biosystems 7900HT Sequence Detection System with FastStart Universal SYBR Green Master mix (Roche Applied Science).

## Western blotting

The standard Western blotting protocol (BioRad) was used with the following specifications. The utricles containing sensory epithelia, transitional epithelia, and underlying mesenchyme were isolated by microdissection and lysed in 50 µL RIPA lysis buffer for 30 min at 4 ˚C and sonicated thrice at low power for 10 s each with the sample kept on ice between the sonications. The total protein concentration in each sample was determined by the BCA assay (Thermo Fisher). A NuPAGE 12% Bis-Tris Protein Gel (Thermo Fisher) was used to resolve the proteins in 5 µg of each sample. The proteins were transferred to a nitrocellulose membrane (BioRad) and blocked for 1 hr at room tempirature in a 5% solution of skim-milk powder (Sigma-Aldrich) in tris buffer (BioRad) with 0.1% Tween 20 (Sigma-Aldrich). After the primary antibodies—rabbit anti-Yap (Cell Signaling) and rabbit anti-H3 (Millipore)—had been reconstituted at 1:10000 in tris buffer blocking solution containing 0.1% Tween 20% and 5% normal sheep serum (Sigma-Aldrich), the membrane was incubated over night at 4 ˚C. After 5 30 min washes at room tempirature in TBST, the anti-rabbit HRP secondary antibody (Millipore) was applied in TBST for 1 hr at room temperature. Horseradish-peroxidase activity was detected with the Amersham ECL Western Blotting System (GE Healthcare Life Sciences).

## Imaging and cell counting

Confocal imaging was conducted with an Olympus IX81 microscope equipped with a Fluoview FV1000 laser-scanning system (Olympus America). To determine the areas of utricular maculae and to enumerate Sox2-positive supporting cells, we imaged wholemounted utricles as Z-stacks and selected representative areas of slices for maximal-intensity Z-projections. To avoid counting Sox2-positive hair cells, Myo7A staining was performed to eliminate the optical sections through the hair cell layer. The statistical significance of comparisons between cell counts was determined by two-tailed Student's *t*-tests. The error bars in figures represent standard errors. *P*-values less than 0.05 are represented by a single star, those less than 0.01 by two stars, and those less than 0.001 by three stars.

For each developmental time point, Sox2-positive cells in the high-resolution image of a maximal-intensity Z-projections of a wholemounted utricle were enumerated automatically with the image-

analysis software CellProfiler (*Carpenter et al., 2006*). We also used this software to automate the areal measurements of utricular sensory epithelia. The algorithms employed for these tasks were tested against manually quantified images and the error of the algorithms was estimated to be consistently lower than 2%.

The intensity of fluorescent staining was measured as mean gray value using ImageJ, which yields a unitless relative value. In a single-channel image, the intensity of a black pixel is 0 and that of a saturated pixel is 255. For RGB images, the maximal intensity of a pixel in each channel is 85 (255/3).

To produce the EdU-intensity plot in *Figure 7D* we used E18.5 utricles labeled for the detection of EdU and Sox2. We then took maximal Z-projections of five EdU-positive images, applied a Gaussian-blur filter to smooth the punctae observed in individual cells, and centered, aligned, and averaged the images. By thresholding and averaging over Sox2-positive channel we calculated the mean outline of the sensory epithelium. To produce the average EdU intensity plots in *Figure 7E*, we performed a spline interpolation of the calculated mean outlines to remove sharp variations. From each image we then copied strips of 400 pixels perpendicular to, and centered on, the smoothed outline. Finally, we calculated the average over all the strips to obtain a plot of the mean EdU density against the distance from the utricle's perimeter.

For the quantifications of cellular densities, EdU labeling, and viral infections at the periphery *versus* center of the utricle and in experiments in vitro, cells were enumerated in areas of 2500–10,000 $\mu m^2$.

## Measurement of elastic moduli

We estimated the elastic modulus of each collagen gel with a piezoelectric bimorph system. An individual gel was mounted between a glass slide and a glass coverslip whose diameter exceeded that of the gel. The mounted gel's original cross-sectional area ($A_0$) and height ($L_0$) were measured. The free end of a piezoelectric bimorph cantilever (PZT-5H, Vernitron Piezoelectric Division, Bedford, OH) mounted on a micromanipulator (MP-285, Sutter Instrument, Novato, CA) was then brought into contact with the glass coverslip. To deliver forces to the gel, the bimorph's base was displaced vertically by the micromanipulator under the control of custom software (*Source code 4*) written in LabVIEW (version 10.0, National Instruments, Austin, TX). The micromanipulator was lowered in 20–50 sucessive steps of constant amplitude, each with a duration of 1 s and with separations of 2 s. The increments for different gels ranged from 0.625 $\mu m$ to 5 $\mu m$. Each displacement induced flexion of the bimorph, resulting in an electrical potential owing to the piezoelectric effect. This signal was amplified and bandpass filtered at 0.1–10 Hz (Grass P55, Astro-Med, Inc., West Warwick, RI). The filtered signal was recorded at sampling intervals of 500 $\mu s$ by a computer running a custom program in LabVIEW.

We calculated the voltage change $\Delta V$ for each force step as the difference between the voltage prior to the pulse and the maximal voltage after the pulse's onset. We next determined from each response both the force delivered by the piezoelectric bimorph and the vertical displacement of the coverslip. For a bimorph cantilever whose free end contacted a glass coverslip, the force $F_B$ generated at the bimorph's free end was $F_B = (K_F W / L) \Delta V$, in which $K_F$ = 2.62 mN/V is the bimorph's sensitivity, $W$ = 6.5 mm is its width, and $L$ = 22 mm is its length (*Corey and Hudspeth, 1980*). The stress applied to the gel was then $\sigma = F_B / A_0$. The displacement $X_B$ of the bimorph's free end was $X_B = F_B / K_B$, in which $K_B$ = 645 N·m$^{-1}$ is the bimorph's stiffness. The difference between the displacement of the bimorph cantilever's base and that of its free end yielded the change in height ($\Delta L$) of the gel and in turn the strain $\varepsilon$ experienced by the gel, $\varepsilon = \Delta L / L_0$.

We estimated the elastic modulus for each gel by fitting the stress-strain relations to the relation $\sigma = \varepsilon E + \sigma_0$, in which $\sigma_0$ is an estimate of the population intercept. All fits possessed coefficients of determination exceeding 0.98. Errors are standard errors of the elastic moduli of multiple gels of a particular composition.

## Numerical simulation of utricular growth

We conducted numerical simulations using CompuCell 3D, a program that allows rapid creation of Glazier–Graner–Hogeweg (GGH) Monte Carlo models (*Swat et al., 2012*). We used a two-dimensional hexagonal lattice and a standard form for the interaction energy $H$ of the system:

$$H = \sum_{\substack{\vec{i},\vec{j} \\ \text{NEIGHBORS}}} J\left[\tau\left(\sigma_{\vec{i}}\right), \tau\left(\sigma_{\vec{j}}\right)\right]\left[1 - \delta\left(\sigma_{\vec{i}}, \sigma_{\vec{j}}\right)\right] +$$

$$\sum_{\sigma} \lambda_{\text{VOL}}[v(\sigma) - V_T]^2 + \sigma \sum \lambda_{\text{SURF}}[s(\sigma) - S_T]^2. \tag{1}$$

The first term on the right describes the contact energy between neighboring cells owing to adhesive interactions. If a cell $\sigma_{\vec{i}}$ of type $\tau\left(\sigma_{\vec{i}}\right)$ occupies the lattice site $\vec{i}$ and a cell $\sigma_{\vec{j}}$ of type $\tau\left(\sigma_{\vec{j}}\right)$ occupies the lattice site $\vec{j}$, then $J\left(\tau\left(\sigma_{\vec{i}}\right), \tau\left(\sigma_{\vec{j}}\right)\right)$ is the boundary energy per unit of contact length. The delta function in the summation indicates that the contact energy between pixels of the same cell is zero.

The second term on the right describes a constraint on the cellular volume, which in two dimensions corresponds to the number of pixels that a cell occupies. A cell $\sigma$ whose volume $v(\sigma)$ deviates from the target value $V_T$ produces an increase in the effective energy proportional to $\lambda_{\text{VOL}}$, the two-dimensional elastic modulus of the cell. This modulus corresponds to the stiffness of a single cell and is distinct from the modulus $E$ of the elastic boundary.

The third sum on the right imposes a constraint on the cell's surface area, which in two dimensions corresponds to the cellular perimeter. Deviations of the surface area from the target value $S_T$ increase the effective energy proportionally to $\lambda_{SURF}$, the elastic modulus of the cell's surface.

Cell dynamics consists of a series of index-copy attempts using a modified Metropolis algorithm. Before each attempt the CompuCell 3D selects a pair of *target* and *source* sites, $\vec{i}$ and $\vec{j}$. If different cells occupy these sites the algorithm then sets $\sigma_{\vec{i}} = \sigma_{\vec{j}}$ with probability given by the Boltzman acceptance function:

$$P\left(\sigma_{\vec{i}} \rightarrow \sigma_{\vec{j}}\right) = \begin{cases} 1 : & \Delta H \leq 0 \\ e^{-\frac{\Delta H}{T_m}} : & \Delta H > 0 \end{cases} \tag{2}$$

in which $\Delta H$ is the change in effective energy, calculated from *Equation 1* and $T_m$ is a parameter describing the amplitude of cell-membrane fluctuations. A Monte Carlo step is defined as $N$ index-copy attempts and sets the computational time unit of the model and, in biologically relevant situations, the step is proportional to experimental time (*Swat et al., 2012*). We converted this measure into actual time by multiplying it by $dt$ and fitting this scaling factor to experimental data. The value of $dt$ depends on the average value of $\Delta H/T_m$. We confirmed by numerical simulations that changing the value of $T_m$ modifies the values of $dt$ and other parameters required to fit the experimental data, but it does not change the result of our simulations in any significant way. We likewise converted areas in pixels to actual areas by multiplying by a factor $dA$ and fitting the results to the experimental data. For further details on CompuCell 3d or the used Metropolis algorithm we refer the reader to *Swat et al. (2012)*.

We studied the dynamics of cellular proliferation and compared two models to explain the observed pattern of growth. $\tau\left(\sigma_{\vec{i}}\right)$ can represent a value either for supporting cells (SC) or for the surrounding elastic boundary (EB). The contact-energy term of *Equation 1* is therefore described by two types of contact energies, $J(\text{EB}, \text{SC})$ and $J(\text{SC}, \text{SC})$. To simplify our notation we define $J(\text{EB}, \text{SC}) = J_{EB}$ and $J(\text{SC}, \text{SC}) = J_C$.

We assumed that the elastic boundary applies hydrostatic compression over the growing sensory epithelium: the stress created by the boundary can change the volume but not the shape of the sensory epithelium. If the elastic boundary is initially unstressed, expansion of the utricle to a surface area $S$ stretches the elastic boundary by $S$ and increases its elastic energy (*Landau and Lifshitz, 1970*) by an amount $H_{ELASTIC} \propto S^2$. In our two-dimensional model this surface area corresponds to the perimeter of the utricle, whereas in three dimensions the relevant value would be the perimeter multiplied by the thickness of the tissue.

The first term of *Equation 1* can be rewritten as

$$\sum_{\substack{\vec{i},\vec{j} \\ \text{BOUNDARY}}} J_{EB}\left[1 - \delta\left(\sigma_{\vec{i}}, \sigma_{\vec{j}}\right)\right] + \sum_{\substack{\vec{i},\vec{j} \\ \text{BULK}}} J_C\left[1 - \delta\left(\sigma_{\vec{i}}, \sigma_{\vec{j}}\right)\right] = H_{BOUNDARY} + H_{BULK}, \tag{3}$$

in which the first sum is evaluated over all the pixels at the boundary of the tissue, for only those contribute to the sum. The second sum is effected for all the pixels in the bulk. The initial sum evaluates as $H_{BOUNDARY} = 2\,J_{EB}S$. Multiplying this term by $S$ reveals that the energy of the elastic boundary is proportional to $S^2$. Identifying $J_{EB}$ with the elastic modulus $E$ of the boundary, **Equation 1** becomes

$$H = \sum_{\substack{\vec{i},\vec{j} \\ \text{BOUNDARY}}} E\left[1 - \delta\left(\sigma_{\vec{i}}, \sigma_{\vec{j}}\right)\right]S + \sum_{\substack{\vec{i},\vec{j} \\ \text{BULK}}} J_C\left[1 - \delta\left(\sigma_{\vec{i}}, \sigma_{\vec{j}}\right)\right] +$$
$$\sum_{\sigma} \lambda_{\text{VOL}}[v(\sigma) - V_T]^2 + \sum_{\sigma} \lambda_{\text{SURF}}[s(\sigma) - S_T]^2 \ . \tag{4}$$

For computational efficiency the sums in the first two terms on the right-hand side of **Equation 3** are evaluated over only a small neighborhood of pixels. This simplification and the discrete nature of the simulations creates artifacts that include sharp corners and anisotropic compression of the tissue when the force produced by the elastic boundary is large.

Cells in our model have a refractory period $T_{ref}$ after division until they can divide again, which represents the time required for a cell to transverce the cell cycle. After this refractory time a cell $\sigma$ divides into two cells when $v(\sigma) \geq V_T/2$. Each cell accordingly grows under the influence of the volume-energy term until it has passed the refractory period and its volume is exceeds the division volume $V_T/2$, whereafter it splits into two cells that resume growth until they again meet the conditions for cell division. The refractory time $T_{ref}$ for each cell is picked from a Gaussian distribution with mean $T_{ref}$ and standard deviation $\sigma_{ref}$.

In our simulations of the elasticity-limited model the rate of cell proliferation decreases to almost zero as the tissue reaches its final size (**Figure 1E**). We observe, though, that the rate of cell divisions is not exactly zero and some residual growth occurs on long timescales. Because of the stochastic nature of our model, random fluctuations eventually push a cell beyond its division volume $V_T/2$ and initiate division. Although the probability for this to happen decreases with the size of the utricle, it is never zero. This behavior poises the utricle in an unstable situation in which, without the presence of an additional mechanism, it grows indefinitely at a very slow pace. We suspect that this instability arises in the utricle and that the Hippo pathway is responsible for completely arresting growth once the utricle has reached its final size. In our model we simulate the action of this pathway in a heuristic way: cells at high density cannot easily increase their volumes and therefore remain unable to divide for a long time. We use this as a measure of density: if a cell is unable to divide after a time $T_{div}$, we render that cell quiescent in our simulation, so that it cannot divide even if it reaches its division volume.

For the elasticity-limited model we adjusted the value of $E$ such that the tissue growth ceased when the number of cells reaches the experimentally observed value. For the morphogen-limited model we considere a much lower value of $E$ such that in the absence of morphogen the sensory epithelium would grow to occupy the entire simulation domain.

The dynamics of the morphogen concentration $[\text{M}]$ is simulated by integrating the diffusion equation

$$\frac{\partial[M]}{\partial t} = D\nabla^2[M] + k_M \sum_{\sigma} \delta[\text{CM}(\sigma)] \tag{5}$$

in which $D$ is the diffusion constant of the morphogen. Each cell $\sigma$ secretes morphogen molecules at a rate $k_M$ from its center of mass $\text{CM}(\sigma)$. When the value of $[M]$ at position $\text{CM}(\sigma)$ reaches a certain threshold $[\text{M}]_{\text{TH}}$ the cell $\sigma$ stops dividing, even if it has reached a volume of $V_T/2$. The integration is conducted with finite differences in space and a forward Euler's scheme in time. During numerical integration of **Equation 5**, time is advanced by $dt$ every Monte Carlo step. CompuCell 3D internally rescales the diffusion equation and adjusts the required number of integration steps per Monte Carlo step to ensure the numerical stability of the method (**Swat et al., 2012**).To simulate EdU pulse chase experiments in the elasticity-limited model (**Figure 6D**) we added a variable $[\text{EdU}]$ to each cell. When the utricle reaches a stage equivalent to E17.5 we activate EdU labeling by setting $[\text{EdU}] = 1$ in each newly divided cell. The time EdU labeling is turned on is adjusted so that the number of labeled cells matches the experiments. After labeling is turned off the concentration of EdU

dilutes by half with every cell division and we assume that it becomes undetectable after four divisions, that is, for $[\mathrm{EdU}] < (1/2)^4$.

Internal cellular pressure is calculated from the model as $P = -2\,\lambda_{\mathrm{VOL}}(v(\sigma) - V_T)$ (*Swat et al., 2012*). Cells with high internal pressure have smaller volumes and therefore higher density. These cells in high density areas are therefore less likely to reach $V_T/2$ and divide.

## Growth of a utricle embedded in an elastic matrix and effective elastic modulus

The utricular macula and non-sensory epithelium form an approximately ellipsoidal chamber filled with endolymphatic fluid and surrounded by mesenchymal stroma and cartilage. The sensory epithelium is located at the bottom of this ellipsoid and is surrounded by band of non-sensory tissue called the transitional epithelium (*Figure 1—figure supplement 1*).

The compressive stress produced by the elastic matrix is transmitted to the epithelial layer as a combination of normal and tangential stresses. Given that the surface upon which the utricular macula lies is relatively flat, we assumed that the normal stress makes little contribution in changing its shape, and therefore could be neglected. This assumption simplified the model's geometry and allowed a two-dimensional representation of the system. The tangential stress is transmitted with the plane of the epithelium. In our two-dimensional representation this stress acted on the boundary of the utricular macula (*Figure 1—figure supplement 1*). We assumed that all the deformations produced in the different tissues are elastic and linearly proportional to the stresses.

Our two-dimensional representation of the utricle comprises a patch of sensory epithelium surrounded by a band of transitional epithelium of Young's modulus $E_1$ and embedded in an elastic matrix of Young's modulus $E_2$ (*Figure 1—figure supplement 1*). We must first demonstrate that these two materials can be described by one band of tissue with an effective Young's modulus $E$ and that the final size of the utricle depends on $E$.

In polar coordinates the band of transitional epithelium has respective inner and outer radii $R_1$ and $R_2$. The elastic matrix has an internal radius of $R_2$; for the sake of simplicity, we assume that it extends to infinity. We designate the displacements of the transitional epithelium and those of the elastic matrix $\vec{u}_1$ and $\vec{u}_2$ respectively. Each of these displacements follows $\nabla \vec{u}_i = c_i$, in which $c_i$ is an arbitrary integration constant (*Landau and Lifshitz, 1970*). Hereafter $i = 1, 2$ represents the variables for the transitional epithelium or elastic matrix respectively. Because the problem is radially symmetrical, only the radial components of the divergence of $\vec{u}$ are nonzero and so we may omit the vector arrow. In polar coordinates the equation reads

$$\frac{1}{r}\frac{\partial(r\,u_i)}{\partial r} = 2a_i. \tag{6}$$

The general solution of this equation is $u_i = a_i\,r + b_i/r$, in which $a_i$ and $b_i$ are integration constants determined by the boundary conditions. The components of the strain tensor are then $u_{rr_i} = a_i - b_i/r^2$, $u_{\phi\phi_i} = a_i + b_i/r^2$, and $u_{zz_i} = 0$. The radial symmetry also implies that the non-diagonal components of the strain tensor are zero.

Using the stress-strain relation for a homogeneous deformation we can write the radial component of the stress tensor as

$$\sigma_{RR_i} = \frac{E_i}{(1+\sigma)(1-2\sigma)}\left[(1-\sigma)u_{RR_i} + \sigma u_{\phi\phi_i}\right], \tag{7}$$

in which $\sigma$ is the Poisson ratio that we assume to be the same for the transitional epithelium and the elastic matrix.

To calculate the different integration constants, we next introduce the boundary conditions of the system. For the elastic matrix, we assume that deformations decay to zero at infinity, that is, $u_2(\infty) = 0$, which implies that $a_2 = 0$. Because there can be no gaps in the boundary between the two regions, the displacements on both sides of the boundary are equal and $u_1(R_2) = u_2(R_2)$. For the system to be in mechanical equilibrium the normal stresses on both sides of the boundary have to be equal, $\sigma_{RR_1}(R2) = \sigma_{RR_2}(R2)$. Finally, we assume that the sensory epithelium generates a hydrostatic pressure $p$ while expanding, so that $\sigma_{RR_1}(R1) = -p$. Using these boundary conditions, we can now solve for the three remaining constants to obtain:

$$a_1 = \frac{-b1 + b2}{R_2^2} \tag{8}$$

with

$$b_1 = \frac{-p\, R_1^2 R_2^2 (1 + \sigma)(E_1 + E_2 - 2E_2\sigma)}{E_1 \left( R_1^2(E_1 - E_2) - R_2^2(E_1 + E_2 - 2E_2\sigma) \right)} \tag{9}$$

and

$$b_2 = \frac{2p\, R_1^2 R_2^2 (1 - \sigma^2)}{R_1^2(E_2 - E_1) - R_2^2(E_1 + E_2 - 2E_2\sigma)}. \tag{10}$$

Inserting these expressions into the equations for $u_i$ and $\sigma_{RR_i}$ yields solutions for the system's deformations and stresses. Eliding this step, we instead show how the combined stiffnesses of the transitional epithelium and elastic matrix can be combined into a single effective term in our simulations.

Following the same steps as before we can calculate the deformation of the effective elastic band of internal radius $R_1$, external radius $R_2$ and Young's modulus $E$ under an internal pressure $p$:

$$u_{eff}(r) = \frac{pR_1^2(1 + \sigma)}{rE}. \tag{11}$$

We next require that the expansion of the sensory epithelium under the effect of this elastic band equals that under the conbined effect of the transitional epithelium and elastic matrix. The change in radius the sensory epithelium is given by the deformation of the inner radius of the transitional epithelium deforms, that is, $R = u_1(R1)$, this relation implies that $u_{eff}(R1) = u_1(R1)$. We can therefore calculate that the effective Young's modulus is:

$$E = \frac{E_1(R_1^2(E_1 - E_2) - R_2^2(E_1 + E_2 - 2E_2\sigma))}{R_1^2(E_1 - E_2)(2\sigma - 1) - R_2^2(E_1 + E_2 - 2E_2\sigma)}. \tag{12}$$

In our bubble-culture experiments we replace the cartilage surrounding the utricle by a collagen matrix. In the mathematical representation of the system this condition is equivalent to a change the value of $E_2$. To more easily see the dependence of $E$ on $E_2$ we can consider a thin band of transitional epithelium and expand *Equation 12* around $R_2 - R_1$ to obtain:

$$E = E_2 - \frac{(E_1 - E2)(E_1 - E_2 - 2E_2\sigma)}{R_1 E_1(\sigma - 1)} \tag{13}$$

from which we can see that the effect of reducing $E_2$ in our experiments can be approximated by a reduction of $E$ in our simulations.

Finally, we can calculate how the radius of the utricle depends on the elastic modulus $E$. The net rate of cell division—the rate of division minus the rate of apoptosis—depends on pressure (*Ranft, 2012*). The homeostatic pressure $P_H$ is the value at which the rates of division and apoptosis are equal and the net division rate is accordingly zero. The disk of tissue grows as long as the pressure exerted by the elastic band is smaller than the homeostatic pressure. Therefore, from *Equation 11* we can see that the total change in radius once the utricle has reached equilibrium is:

$$\Delta R = u_{eff}(R_1) = \frac{P_H R_1(1 + \sigma)}{E}. \tag{14}$$

The change in radius is proportional to $1/E$, and therefore the change in area of the utricle is proportional to $1/E^2$. The equilibrium areas of simulated utricles for different values of the Young's modulus agree with this expectation (*Figure 1F*).

## Acknowledgements

The authors thank members of their research group and Dr. N Segil for comments on the manuscript. KG was supported by National Institute on Deafness and Other Communication Disorders; AJ

and AP were supported by FM Kirby Foundation; JS was supported by National Institute on Deafness and Other Communication Disorders and by National Institute of General Medical Sciences. All authors were also supported by Howard Hughes Medical Institute, of which AJH is an Investigator.

## Additional information

### Funding

| Funder | Grant reference number | Author |
| --- | --- | --- |
| Howard Hughes Medical Institute | | A J Hudspeth |
| Robertson Therapeutic Development Fund | | Ksenia Gnedeva A J Hudspeth |
| National Institute on Deafness and Other Communication Disorders | 2T32DC009975-07 | Ksenia Gnedeva |
| F. M. Kirby Foundation | DSG2000164 | Adrian Jacobo |
| National Institute on Deafness and Other Communication Disorders | F30DC013468 | Joshua D Salvi |
| National Institute of General Medical Sciences | T32GM007739 | Joshua D Salvi |

The funders had no role in study design, data collection and interpretation, or the decision to submit the work for publication.

### Author contributions

KG, Conceptualization, Data curation, Formal analysis, Supervision, Funding acquisition, Validation, Investigation, Visualization, Methodology, Writing—original draft, Writing—review and editing; AJ, Conceptualization, Data curation, Software, Formal analysis, Validation, Investigation, Visualization, Methodology, Writing—original draft, Writing—review and editing; JDS, Data curation, Software, Formal analysis, Validation, Investigation, Visualization, Methodology, Writing—original draft; AAP, Investigation, Writing—original draft; AJH, Conceptualization, Resources, Supervision, Funding acquisition, Validation, Visualization, Methodology, Writing—original draft, Project administration, Writing—review and editing

### Author ORCIDs

Ksenia Gnedeva, http://orcid.org/0000-0002-3870-9256
Adrian Jacobo, http://orcid.org/0000-0001-9381-6292
Joshua D Salvi, http://orcid.org/0000-0002-5140-7746
A J Hudspeth, http://orcid.org/0000-0002-0295-1323

### Ethics

Animal experimentation: Experiments were conducted in accordance with the policies of The Rockefeller University's Institutional Animal Care and Use Committee (IACUC Protocol 15832) and the Keck School of Medicine of the University of Southern California (IACUC Protocol 20108).

## Additional files

### Supplementary files

• Source code 1. CompuCell 3D (*Swat et al., 2012*) scripts and parameter files for the elasticity-limited model.

• Source code 2. CompuCell 3D (*Swat et al., 2012*) scripts and parameter files for the morphogen-limited model.

• Source code 3. CompuCell 3D (*Swat et al., 2012*) scripts and parameter files for the stem cell model.

• Source code 4. LabVIEW scripts to control the MP-285 micromanipulator for measuring the stiffness of collagen gels.

## Major datasets

The following previously published dataset was used:

| Author(s) | Year | Dataset title | Dataset URL | Database, license, and accessibility information |
|---|---|---|---|---|
| Gnedeva K, Hudspeth AJ | 2015 | Gene expression in the developing murine utricle | https://www.ncbi.nlm.nih.gov/geo/query/acc.cgi?acc=GSE72293 | Publicly available at the NCBI Gene Expression Omnibus (accession no: GSE72293) |

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
