## [Decision Letter]

Thank you for submitting your article "Mechanical force restricts the growth of the murine utricle" for consideration by *eLife*. Your article has been reviewed by three peer reviewers, and the evaluation has been overseen by Tanya Whitfield as the Reviewing Editor and Andrew King as the Senior Editor. The following individuals involved in review of your submission have agreed to reveal their identity: Alexander Fletcher (Reviewer #1) and Jennifer Stone (Reviewer #3).

The reviewers have discussed the reviews with one another and the Reviewing Editor has drafted this decision to help you prepare a revised submission.

Gnedeva et al. address the very interesting question of the mechanisms responsible for growth arrest during development, in particular the role that mechanical constraint plays in this process. This question is especially pertinent concerning the mammalian inner ear, since growth arrest might be linked to the inability of mammals to produce new sensory hair cells in adults, making hearing and balance dysfunctions often irreversible.

This work combines analysis of the murine utricle in vivo and in a novel ex vivo culture system with computational modelling to address the mechanisms underlying the abrupt growth arrest of this tissue observed late in embryonic development. The authors conclude that, as the utricle grows, it experiences a resistive force from the surrounding elastic environment. This force creates a cell-density gradient across the tissue that is interpreted through the subcellular localization of Yap, and corresponding gradient of supporting-cell proliferation. The authors demonstrate that this hypothesis can explain the observed tissue growth dynamics when the stiffness of the surrounding collagen gel is reduced. These results reflect a tissue-level coordination of proliferation based on mechanical conditions at the boundary, and are not consistent with a model where growth is driven by stem cells located at the centre of the utricle.

Essential revisions:

1) Please incorporate the extra details requested by reviewer 1 into the model.

2) Further experimental work is needed to support the conclusions concerning the regeneration post-injury in both soft and stiff gels.

a) The role of Yap should be shown more clearly both during regeneration upon injury of the utricle and in the doubling of the utricle grown in a softer gel. Please see comments from reviewer 2.

b) The authors should determine if dividing cells are indeed Sox2^+^ (sensory epithelial) cells and if they give rise to hair cells, in order to determine whether their observations are relevant to hair cell production. This could be done with pulse/fix experiments for Sox2 and pulse/chase experiments for identification of post-M cells. Please see comments from reviewer 3.

3) Reviewer 3 has made a number of suggestions for additional literature that could be cited to acknowledge previous relevant work. Please take this into account.

4) It is essential that all methods are described in sufficient detail to allow others to repeat the study. Please see the comments from reviewer 3.

5) All three reviewers have suggested numerous small changes or corrections that would help to clarify or improve the manuscript. Please attend to these where possible.

The full reviews are appended below for further information.

Reviewer #1:

Overall, I found the manuscript to be clearly written, well motivated, and an important contribution to our understanding of the role of mechanics in tissue growth and size control. As such, I do not have any explicit requests for additional work, though I do have a number of queries and concerns that I would like to see addressed or clarified in the manuscript.

There are a few details missing from the computational model that prevent a full critique of the results. First, in subsection “Numerical simulation of utricular growth and cellular differentiation” the authors refer to "the surrounding elastic boundary", but based on Figure 2 it seems that there is also an elastic collagen gel underlying the cells. Thus, for a stiffer gel, cell movement in the centre of the tissue (not just at the periphery) should be affected; could this affect the relative timescales of mechanical relaxation and cell growth, and thus potentially affect the observed growth dynamics? Second, the authors assume that the link between mechanical force and cell-cycle arrest occurs specifically via the pressure (compression) on each cell, but it was not immediately clear to me that this must be the way that mechanosensing occurs in this system; has this been demonstrated experimentally?

A further assumption is that the growing utricle is essentially two-dimensional (Results section); yet the authors state in subsection “Dissection and culturing of utricles” that the utricular bubble is "a spherical structure". It was not quite clear to me how good the planar approximation is for this tissue, and whether tissue curvature could complicate the mechanical response to a stiff surrounding medium.

In addition, in subsection “Dynamics of utricular development” the authors refer to "processes such as cell division, differentiation, molecular signaling, and physical force", but do not refer to apoptosis or extrusion from the utricle. Since they do refer to apoptosis in subsection “Growth of a utricle embedded in an elastic matrix”, it would be helpful to know whether any cell death is observed during the experimental period in the ex vivo culture system. In other tissues, apoptosis and live-cell extrusion have been demonstrated to locally relieve stress, thus contributing to mechanical homeostasis in the tissue (e.g. Eisenhoffer et al., doi:10.1016/j.tcb.2012.11.006). It would therefore be of interest to know whether apoptosis occurs in the utricle, since this could affect the authors' mechanical argument.

I cannot comment with authority on the experimental protocols, but was unsure whether one might expect any matrix or collagen remodelling; is it reasonable to assume no mechanical feedback from the utricle back on the surrounding medium over the timescale of interest? Also, the authors use the phrase "mechanical force" in the Title and Abstract, but this is quite a general term, and could be made more precise – the authors' conclusion is that it is in particular the resistive, elastic force from the surrounding medium that drives growth arrest.

Reviewer #2:

Overall this is a nice study and the paper is clearly written and pleasant to read. Yet I think some points need to be addressed or clarified.

1) Figure 3: The authors have shown in Figure 1 that the utricule area increases rapidly between E15.5 and P2 after which growth slows down and stops. Why then looking at Yap protein level only at P14. If a decrease in Yap activity accounts for this growth arrest, one would expect a decrease in Yap protein and activity as early as P2. Same applies to the expression of Yap target genes and genes encoding inhibitors of Yap translocations.

2) Figure 4: The authors show an increase of proliferation and of Yap protein during regeneration upon injury of the utricule. These are only correlative observations and no proof that the observed proliferation depends on Yap, nor that Yap plays a role in regeneration.a) The authors should show, using the same canonical Hippo target genes they have used in Figure 3 that Yap activity is indeed increased during regenerationb) The authors should perform this experiment while blocking Yap function using for example the blocking peptide to show that regeneration is then blocked

3) Figure 5: Some controls and a more in-depth analysis are required to convincingly show that Yap activity is solely responsible for increase proliferation and a doubling of the utricule in a less stiff gel. Also here,a) The authors should show, using the same canonical Hippo target genes they have used in Figure 3 that Yap activity is indeed increased in a softer gel as compared to a stiff gelb) The authors should use an alternative method, next to the blocking peptide, at best a yap1 mutant, to show that utricule area and proliferation also do not increase in a soft gel.

4) Figure 5 text: The last paragraph of subsection “Yap is necessary for supporting-cell proliferation”

"Unexpectedly, the supporting-cell density in GFP-YTIP-infected utricles was significantly lower than that in GFP-infected control cultures (Figure 5). As a result, although the area doubled, the total number of supporting cells in GFP-infected cultures increased only 20% in comparison with GFP-YTIP cultures (Figure 5). " This is very unclear. "Lower" should be "higher". The fact that "the total number of supporting cells in GFP-infected cultures increased only 20% in comparison with GFP-YTIP cultures" is to my understanding the cause and not the consequence of the higher density.

"These data suggest that Yap activity is also required for cell spreading. Additional experiments are necessary to validate this prediction and to uncover the underlying mechanism." I think this statement is fully in the scope of this paper and the authors should show this increased spreading by measuring the cell area and the distance between individual cells.

5) Figure 5—figure supplement 1: The authors should clarify in which panel of B, one looks at an apical or at a basal view. More importantly, hair cells in the utricule in 40Pa gel appear much bigger than in a 640Pa gel. The authors do not mention this observation. Given that Yap is not expressed in hair cells, how do the authors explain this?

6) Figure 6: It is not clear to me how the cell density can increase so much between E15.5 and E17.5 in vivo (Figure 6) if both the supporting cell number (Figure 6) and the utricule area (Figure 1) increase at about the same speed. In this case, I would expect the cell density not to change much.

7) a) It is not clear whether the authors interpretation is that both the decrease in cell density (or increased spreading) and increase in proliferation are driven by increased Yap activity or whether the decrease in stiffness results in decrease in cell density (yap independently) and the decrease in cell density, in turn, drives yap-dependent proliferation. This should be clarified in a precisely justified manner by the authors.b) The initial question the authors stated in the Abstract and in the Introduction is concerning the link between inner ear growth arrest and inability to regenerate hair cells. Although the authors show that support for cell proliferation can be enhanced by lowering the elastic mechanical forces of the surrounding tissue, they do not show the consequences on hair cell regeneration. It would be very interesting that the authors damage the hair cells of ex-vivo cultured utricules in both 40Pa and 640Pa gels to compare their capacity to regenerate hair cells.

Reviewer #3:

This is a beautiful paper examining something quite novel and significant – the effect of organ stiffness and epithelial cell density on the developmental growth of the sensory epithelium for a balance organ, the utricle. Overall, the paper is well composed, and the data that are presented are compelling and easy to understand. The role of stiffness on inner ear organ development has been understudied, and this is the first report to my knowledge of the role of Hippo signaling in controlling addition of new cells to a growing organ in the inner ear. Unfortunately, the paper needs quite a bit of editing to make it suitable for publication, as several statements are vague or misleading, and some aspects of the data are hard to interpret because they are incomplete or presented in an unclear or in one case incorrect manner. Further, many methods were not described.

The lack of line numbers and labels on the figure made it particularly challenging to review this proposal.

Below, I summarize the major concerns for each section.

Abstract

Please take into consideration that several labs (Forge, Raphael, Stone labs) have shown there is some spontaneous replacement of vestibular hair cells in adult rodents, probably due to supporting cell to hair cell conversion, and perhaps even in humans (Taylor et al., 2015). Plus, Warchol et al., (1993) showed that cultured adult utricles from guinea pigs and humans have considerable cell division in the macula (presumably of supporting cells) and Rubel et al., (1995) showed cell division leading to supporting cell replacement occurs in adult rodent vestibular epithelia.

Add "the" in between "of" and "utricle's development".

Introduction

The statement that hair cells almost never contact one another is misleading, especially since a recent paper (Pujol et al., 2014) showed that in the adult mouse utricle, a large proportion of the type II hair cell population is in contact via basolateral processes.

In the fourth sentence the statements are untrue or misleading and should be modified. They suggest that supporting cell division does not proceed into the postnatal period and vestibular hair cells are not replaced after birth. However, Ruben (1967) showed that significant numbers of new vestibular epithelial cells are made postnatally; this paper should be cited. Further, Burns et al., (2012) showed continued macular cell proliferation into the neonatal period and production of several hair cells from the products of those dividing cells. The wording suggests that the authors have not read several papers showing that there is considerable regeneration of vestibular hair cells in adult rodents, probably by direct transdifferentiation. By these accounts, percentages of regenerated hair cells range from 20% to 70% of normal levels (in some regions). Several references should be cited, including Forge et al., 1993; Forge et al., 1998; Kawamoto et al., 2009; Lin et al., 2011; Golub et al., 2012; and Slowik et al., 2013. Further, these findings should be taken into consideration and addressed in the Discussion in the context of this paper's findings during development.

Second paragraph, third sentence. Since White et al., (2012) saw the EGF effect in chicken auditory organs, I do not believe this sentence helps. Several groups have examined EGF signaling in adult mouse utricles (e.g., Oesterle's group and Gao's group). Although White et al., assessed effects of EGF on mouse cochlear supporting cells, they were dissociated cells, not intact epithelia.

I would consider mentioning here work from the Pirvola and the Burns groups on the failure of cell cycle effectors such as cyclin D to promote division in mouse utricles. This seems quite germane to this discussion.

Results section. Why assume supporting cells secrete a growth-inhibiting molecule? Why not hair cells, or why couldn't supporting cells secrete a growth-promoting molecule? Would the observations still rule out a morphogen model if the morphogen were growth-promoting?

Figure 7 disagree on several points here; what is stated does not match the figure. First, cytoplasmic Yap labeling looks strong in the images. I think it is clear from the images in Figure 3 that no nuclear Yap is seen in the central region while some nuclear Yap is seen in the periphery. It is clear however, that some cytoplasmic Yap is retained in the periphery, which makes it a bit hard to discern.

Subsection “Yap signaling during development and regeneration of the murine utr”. Need to add Kawamoto et al., 2009 to Golub et al., 2012 to reference list.

Subsection “Yap signaling during development and regeneration of the murine utr”, Figure 4. This is an odd experiment to perform, as opposed to targeted hair cell ablation, which could be accomplished with neomycin in vitro using published methods. One concern is that the authors are not actually examining cell divisions related to sensory epithelial cell repair. Were the EdU^+^ cells Sox2^+^? This would indicate if the cells dividing were indeed supporting cells or some other cell, such as from the stroma or the transitional epithelium. Did the authors investigate the fate of the post-M cells in that region? This would indicate divisions are leading to hair cells. I see this as a major problem with interpreting the data.

Subsection “Yap is necessary for supporting-cell proliferation”. No – the density was higher! Did you mean macular area? Overall, this section is confusing. Here is what the data in Figure 5 say to me: In YTIP-treated epithelia, macular area was halved (E) and dividing cell numbers were reduced to ~20% of normal (H) while supporting cell density doubled (G). This is not surprising to me, but I don't understand why this suggests that cell spreading was reduced. Further, it is strange to me is that epithelial cell numbers were not dramatically reduced, given the EdU data.

Subsection “Mechanical force confines cellular proliferation to the utricle’s periphery”. Utricular growth does not cease at P2. Burns et al., 2012 (Figure 1) show macular area increases significantly out to P8.

Discussion section This is a short Discussion. The authors should consider addressing the following as well:

Does Yap play a similar role in limiting growth of any other sensory or neural tissues (retina, olfactory epithelium, other)? It would be a good addition to discuss this here, briefly.

Is the favoured model consistent with all forces controlling cell division being restricted to the sensory epithelium (i.e., related to epithelial cell density)? Might some forces be derived from outside the macula? If so, what could they be?

What significance is the boundary of transitional epithelial (TE) cells with respect to limiting supporting cell division and growth? What is Yap expression like in the TE?

Why might supporting cell division eventually cease? What changes in the stiffness of cell density might allow this to occur?

Do you feel you have ruled out the possibility that both types of forces – morphogen and elasticity – might be actively controlling utricular size?

During damage, how might extrusion or degradation of hair cells alter Hippo signaling?

Video 3: This seems like new data; why isn't it included in the Results section?

Methods

No methods were provided for the following: Western blots, RNAseq, virus experiments! For the Western and RNAseq, were sensory epithelia only used or were whole utricles used?

Subsection “Mechanical force confines cellular proliferation to the utricle’s periphery”:

– What% of the macula was sampled?

– Were cells from each zone (striola, lateral extrastriola, medial extrastriola) sampled?

– In utricles, type II hair cells are also *Sox2Sox2*^+^; how did you assure you were not counting *Sox2Sox2*^+^ hair cells? (This is particularly an issue with manual counting methods).

– What was the degree of error with automated areal measurements? (This should be known, since the algorithms were compared to manual counts).

Subsection “Mechanical force confines cellular proliferation to the utricle’s periphery”. There is no Figure 4. What does this mean: "calculated the mean outline?" Is this area?

Subsection “Mechanical force confines cellular proliferation to the utricle’s periphery”. There is no Figure 4. Since I cannot link this information to any figure, I cannot interpret or assess this paragraph.

Figure 2, here, and for several places throughout the manuscript: N's should be provided for each group (e.g., N=3 for 40 Pa and N=3 for 640 Pa)

– Figure 2 source data file is very confusing. I opened the files in Excel. First, it states supporting cell area instead of epithelial area. Second, it is not clear which group (40, PA 640 PA or in vivo) is described in each column, since there seem to be column titles missing for each column. Column B has no heading at all.

– Figure 2 source data. Again, I cannot see the column titles, so I am not sure they are there. I recommend carefully checking all source data to make sure they are complete (have all headings) and match the terms used in the figures and text.

Figure 3, are you confident the cells with the higher Yap intensity are in the sensory epithelium, or might they be transitional epithelial cells? What are the units for fluorescence intensity?

– It is impossible to tell that hair cells do not express Yap. The reader would be more convinced if fluorescence were shown to not overlap with cytoplasmic hair cell marker labelling. Another possibility is to reference Figure 5—figure supplement 1, since it is quite clear there that hair cell cytoplasm isn't labelled.

Figure 4 legend, regarding the statement "The same cells enter the cell cycle". This is true for some cells, but lots of dividing cells seem NOT to have translocated nuclear Yap. This could be misleading or suggest a bias. These images should be shown at higher magnification, to better enable readers to see the Yap and EdU labelling in the nuclei.

Figure 5, did the authors examine the cell-autonomous nature of the effect of GFP-YTIP? Do they know if there was a significant change in the GFP+ versus GFP- cells? This would be very important to understand, as it could expand our understanding of how Yap functions.

– I think it's important to note that the supporting cell density decreased with low stiffness (Figure 5). How about effects on total supporting cell numbers?

Figure 6 do not understand what is being shown here. What is the dotted line? I do not see any shaded area around the line. What does x axis represent?

– Figure 6, how were the center and the periphery defined for these counts? How was this unbiased? Are data in Figure 6, and I from real utricles or simulated data?

– The authors should clearly state which cells were counted by including "supporting" or "macular" in the y-axis labels whenever possible.

– What is significance of asterisks in the graphs? This should be explained in the legend.

[Editors' note: further revisions were requested prior to acceptance, as described below.]

Thank you for resubmitting your work entitled "Elastic force restricts the growth of the murine utricle" for further consideration at *eLife*. Your revised article has been favourably evaluated by Andrew King (Senior editor), a Reviewing editor (Tanya Whitfield), and three reviewers.

The manuscript has been substantially improved and reviewers 1 and 2 are happy that their queries and concerns have been addressed. Reviewer 3 is very positive overall but has some remaining suggestions concerning wording or comparison to published studies. We are therefore returning the manuscript to give you the option of addressing or rebutting these concerns, which should be quick to do. The detailed comments are included below.

Reviewer #3:

This excellent paper has been significantly enhanced by the added experiments and the careful editing of the text. I have only a few additional suggestions before publication.

Abstract. I recommend rephrasing the first sentence to "limited ability of cells in the inner ear to proliferate", as inner ear proliferation would suggest duplications of the organ.

Introduction. I am in favor of the wording now, with two small gripes.

The Warchol et al., 1993 Science paper refutes the last statement that "supporting cells [...] lose the ability to reenter the cell cycle after neonatal stages", since they showed that cultured utricles from adult humans and guinea pigs attained numerous tritiated thymidine-labeled supporting cells over time. I realize this is a stand-out paper, but I wonder if you could add to the reference list, "but see Warchol et al., 1993". This applies to text in other places in the manuscript (e.g., subsection “Yap signaling during development and regeneration of the murine utricle”).

I recommend switching the last sentence, which reads "making hearing and balance loss" to "and may be a primary reason why hearing and balance function fail to recover in mammals after hair cell damage". In my opinion, it is too bold to suggest that the lack of proliferation is THE cause for the lack of functional recovery.

Subsection” Yap signaling during development and regeneration of the murine utricle”. Please remove Kawamoto et al. and Golub et al., from this list; only experiments on adult mice were performed in these studies.

Subsection “The effect of elastic force on the pattern of supporting-cell proliferation”. I originally disagreed with the statement that utricular growth ceases at P2, and I still find information confusing, since no changes were made to the text.

It's important to distinguish between cell division and organ growth. Although cell division may cease at P2, the organ could continue to grow beyond that point. Indeed, Burns et al., 2012b found evidence to support that the utricle does grow in size and hair cell numbers increase after P2. I recommend removing the reference to Burns et al., 2012b here, or providing a clearer explanation of the disparate conclusions in the two papers, as you did in your response to reviewers.

I will also note that Figure 1 of Burns et al., 2012a and Figure 4 of Burns et al., 2012b show BrdU incorporation in utricular macula cells after P2. So, I am also confused by the declarations throughout that supporting cell proliferation ceases within the first 2 days of postnatal life.

These may seem like petty gripes, but it's important to be accurate, and I feel some findings are stated as absolutes that are misleading.

---

## [Author Response]

*Essential revisions:*

*1) Please incorporate the extra details requested by reviewer 1 into the model.*

Following reviewer 1’s suggestions, we have expanded the section “Dynamics of Utricular Development” and the Methods. We have added a supplementary figure (Figure 1—figure supplement 1) to better explain the geometry of the utricle and the simplifications that we have made to translate this geometry into our two-dimensional model. We have also rewritten the section “Growth of a utricle embedded in an elastic matrix and effective elastic modulus” in the Methods to including additional calculations, showing explicitly how the translational epithelium and surrounding tissue can be combined into an effective description described by a single Young’s modulus.

We have also included additional details about the Monte Carlo simulations and the diffusion equation for the growth-inhibiting molecule.

*2) Further experimental work is needed to support the conclusions concerning the regeneration post-injury in both soft and stiff gels.*

*a) The role of Yap should be shown more clearly both during regeneration upon injury of the utricle and in the doubling of the utricle grown in a softer gel. Please see comments from reviewer 2.*

As suggested by reviewer 2, we have conducted additional experiments to demonstrate the role of Yap in supporting-cell proliferation more clearly. The summaries of two major experiments are provided below.

We have performed injury experiments while blocking nuclear Yap function with YapTead interfering peptide (YTIP). In support of the role of Yap in cell-cycle reentry during regeneration, these experiments demonstrate that the number of Sox2^+^ supporting cells that undergo mitosis is reduced significantly in GFP-YTIP cultures as compared to GFP controls. These data have been incorporated into Figure 4 and the supplementary file – Figure 4 – has been uploaded with the current submission.

To demonstrate nuclear Yap signaling activation, we have additionally performed gene-expression analysis on utricles cultured in low-stiffness and high-stiffness gels. Quantitative PCR demonstrates that canonical nuclear Yap target genes are significantly upregulated in the utricles cultured in low-stiffness gels as compared to high-stiffness controls. These data have been incorporated into Figure 5 supplementary file with corresponding name has been uploaded with the current submission.

*b) The authors should determine if dividing cells are indeed Sox2^+^ (sensory epithelial) cells and if they give rise to hair cells, in order to determine whether their observations are relevant to hair cell production. This could be done with pulse/fix experiments for Sox2 and pulse/chase experiments for identification of post-M cells. Please see comments from reviewer 3.*

As suggested by reviewer 3, we have added high-magnification images that clearly demonstrate that the cells at the injury site that translocate Yap to their nuclei and enter the cell cycle are Sox2^+^ supporting cells. These data have been incorporated into Figure 4.

In Figure 2 and Figure 6, we demonstrate that cell proliferation is, in fact, seen in Sox2^+^ supporting cells in 40 Pa gel cultures. In addition to these data, we have now provided Sox2/Yap/EdU triple labeling in utricular bubbles after 4 days in culture in Figure 5—figure supplement 1. These data confirm that in 640 Pa gels, similar to the observation in vivo, proliferation of Sox2^+^ supporting cells is seen only at the macular periphery, where Yap labeling can be seen in the nuclei. In 40 Pa gels, by contrast, nuclear Yap labeling and EdU incorporation are observed throughout the sensory epithelium of the utricle. We believe that these data, in conjunction with Yap loss of function experiments and the analysis of Yap downstream signaling activation, convincingly demonstrate the key role of nuclear Yap in supporting-cell proliferation.

We wish to clarify that this work’s focus is on the role of elastic forces and Yap signaling in supporting-cell proliferation both, during development and during cell-cycle re–entry after injury. Although it would be very interesting to look at the sensory receptor regeneration, to our knowledge, *proliferative* hair-cell regeneration is not typically observed in vitro. Supporting cells in the utricle of a young adult mouse retain a limited capacity to re-enter the cell cycle after hair cells have been lost (Burns et al., 2012a; Wang et al., 2015). After mitosis, however, these cells do not give rise to new sensory receptors in culture. The EdU pulse-chase experiments in the papers provided above show that new hair cells, formed in utricular cultures after hair-cell damage, come from direct conversion of non–proliferative supporting cells. We therefore believe that experiments demonstrating the fate of post-mitotic supporting cells, re-entering the cell cycle after an injury or when the elastic force is reduced, must be executed in vivo and accordingly lie beyond the scope of the current work

*3) Reviewer 3 has made a number of suggestions for additional literature that could be cited to acknowledge previous relevant work. Please take this into account.*

We have rewritten the Introduction to include most of the citations recommended by the reviewer.

*4) It is essential that all methods are described in sufficient detail to allow others to repeat the study. Please see the comments from reviewer 3.*

We have added the missing methods (e.g. Western Blots, qPCR, viral production) to the manuscript. As suggested by the third reviewer, we have also updated the labels on all the supplementary data files, which have been cited in the corresponding figure legends and provide transparent primary data in support of our quantifications and statistical analyses.

*5) All three reviewers have suggested numerous small changes or corrections that would help to clarify or improve the manuscript. Please attend to these where possible.*

We have taken all of the reviewers' comments into a careful consideration and incorporated the corresponding corrections into the revision. Our detailed responses to reviewers can be found below.

*The full reviews are appended below for further information.*

*Reviewer #1:*

*Overall, I found the manuscript to be clearly written, well motivated, and an important contribution to our understanding of the role of mechanics in tissue growth and size control. As such, I do not have any explicit requests for additional work, though I do have a number of queries and concerns that I would like to see addressed or clarified in the manuscript.*

We thank the reviewer for his positive comments and constructive critique of our manuscript. We did our best to clarify his points and we hope that our theoretical approach is now clearer as a result.

*There are a few details missing from the computational model that prevent a full critique of the results. First, in subsection “Numerical simulation of utricular growth and cellular differentiation” the authors refer to "the surrounding elastic boundary", but based on Figure 2 it seems that there is also an elastic collagen gel underlying the cells. Thus, for a stiffer gel, cell movement in the centre of the tissue (not just at the periphery) should be affected; could this affect the relative timescales of mechanical relaxation and cell growth, and thus potentially affect the observed growth dynamics?*

In the section “Dynamics of Utricular Development” we have included a more detailed explanation of the geometry of the utricle and how it can be represented in a two–dimensional model. We have included a new Figure 1—figure supplement 1 to clarify this. We have also modified the section “Growth of a utricle embedded in an elastic matrix” in the Methods to show how the Young’s modulus of the surrounding cartilage and mesenchymal stroma can be incorporated in the two-dimensional model as an effective Young’s modulus of the surrounding elastic boundary. In the model all the stresses exerted by the surrounding cartilage or collagen matrix are applied to the utricular macula at its boundary, through the surrounding band of transitional epithelia. Stresses then propagate to the center of the sensory epithelium affecting the motility and growth of cells at this location.

The replacement of cartilage by collagen in the experiments has the effect of changing the effective Young’s modulus of the elastic boundary and, in our two-dimensional description, this accounts for all the stiffness changes in the system.

We wish to note that in these experiments the collagen matrix is *not* in direct contact with the sensory epithelium, but is separated by stromal cells. Therefore, on a more refined three-dimensional model of the system one would have to incorporate a friction coefficient between stromal and sensory cells. Our two-dimensional approximation allows us to simplify these details and incorporate their contributions into the effective stiffness of the elastic boundary.

*Second, the authors assume that the link between mechanical force and cell-cycle arrest occurs specifically via the pressure (compression) on each cell, but it was not immediately clear to me that this must be the way that mechanosensing occurs in this system; has this been demonstrated experimentally?*

In the context of tumor growth it has been shown that mechanical forces can impair cell proliferation (Helmlinger et al., 1997; Cheng et al., 2009; Montel et al., 2011). Furthermore, a link between stretching and compressive forces has been suggested to act as part of the size-control mechanism of the *Drosophila* wing disk (Hariharan, 2015), but the mechanisms coupling external force, the Yap signaling pathway, and cell-cycle arrest are not completely understood. A variety of mechanical cues, such as cell-cell contact, cellular density, stretching, and cellular tension, have recently been identified as regulators of Yap activity (Gaspar et al., 2014; Low et al., 2014).

Although the internal pressure of cells is difficult to demonstrate experimentally, we assume that compressive forces regulate cell-cycle arrest in our system because of our model's predictions and experimental results. In Figure 7, we have demonstrated that, according to the model, the cells at the center of the utricle have higher internal pressure, meaning these cells are under stronger compressive forces. Our experimental data support that prediction, for the cellular density at the center of the utricle is in fact greater than that at the organ’s periphery. Further evidence comes from experiments in three-dimensional organotypic cultures (Figure 5). We believe these data demonstrate that decreasing the stiffness of the gels relieves compression, allowing cells at the center of the utricle to expand in size and proliferate. We think that the decrease in cell density observed in these experiments results directly from decreased compression. As a result, Yap is translocated into the nuclei and cells re–enter the cell cycle.

*A further assumption is that the growing utricle is essentially two-dimensional (Results section); yet the authors state in subsection “Dissection and culturing of utricles” that the utricular bubble is "a spherical structure". It was not quite clear to me how good the planar approximation is for this tissue, and whether tissue curvature could complicate the mechanical response to a stiff surrounding medium.*

We apologize for the lack of precision in the description of the tissue. Either an intact utricle or a utricular bubble has an ellipsoidal shape with the sensory macula at its bottom. We have modified the text to reflect this. As we mentioned before, we have also expanded the text to explain the geometry of the organ in better detail and added a supplementary figure to clarify this point.

As can be seen in the new Figure 1—figure supplement 1, the radius of curvature of the sensory epithelium is much larger than the typical length scales along the surface of the utricular macula. Because the curvature is small compared to the other relevant length scales in the system, we have approximated the sensory epithelium as a two–dimensional surface. This approximation simplifies our theoretical model and dramatically reduces computation times.

*In addition, in subsection “Dynamics of utricular development” the authors refer to "processes such as cell division, differentiation, molecular signaling, and physical force", but do not refer to apoptosis or extrusion from the utricle. Since they do refer to apoptosis in subsection “Growth of a utricle embedded in an elastic matrix”, it would be helpful to know whether any cell death is observed during the experimental period in the* ex vivo *culture system. In other tissues, apoptosis and live-cell extrusion have been demonstrated to locally relieve stress, thus contributing to mechanical homeostasis in the tissue (e.g. Eisenhoffer et al., doi:10.1016/j.tcb.2012.11.006). It would therefore be of interest to know whether apoptosis occurs in the utricle, since this could affect the authors' mechanical argument.*

During the developmental stages considered in this paper, apoptosis does not make a major contribution to the mechanics of size regulation. We have stained utricles for activated caspase 3, a marker for apoptosis, and have seen only a few labeled cells in the utricular macula. Our interpretation of this result is that apoptosis of a few scattered cells can relieve local stresses in the utricle but has no impact in setting its final size. We are therefore confident about the appropriateness of neglecting the contribution of apoptosis in the present version of our model.

*I cannot comment with authority on the experimental protocols, but was unsure whether one might expect any matrix or collagen remodelling; is it reasonable to assume no mechanical feedback from the utricle back on the surrounding medium over the timescale of interest? Also, the authors use the phrase "mechanical force" in the Title and Abstract, but this is quite a general term, and could be made more precise – the authors' conclusion is that it is in particular the resistive, elastic force from the surrounding medium that drives growth arrest.*

The utricular sensory epithelim, which contains the supporting cells and hair cells that we study, never comes into direct contact with the three-dimensional collagen matrix. Instead, the mesenchymal stroma that normally surrounds the inner-ear organs attaches the utricle to the collagen fibers. Matrix remodeling therefore should not affect the sensory epithelium, for the basal lamina (basement membrane) that separates the epithelium from the mesenchyme, remains intact in bubble cultures. Moreover, because the volume of collagen matrix is much greater than that of the utricle – the diameter of the utricular macula is around 0.15 mm and the volume of the collagen matrix is 450 mL – we assume that any mechanical feedback produced by the utricle onto the collagen can be neglected. Finally, we have updated the language in the Title and Abstract to explicitly indicate that the force is elastic in nature.

*Reviewer #2:*

*1) Figure 3: The authors have shown in Figure 1 that the utricule area increases rapidly between E15.5 and P2 after which growth slows down and stops. Why then looking at Yap protein level only at P14. If a decrease in Yap activity accounts for this growth arrest, one would expect a decrease in Yap protein and activity as early as P2. Same applies to the expression of Yap target genes and genes encoding inhibitors of Yap translocations.*

We understand the reviewer’s point and wish to clarify our decision for selecting the later time points for some of the analysis. We think that loss of nuclear Yap signaling owing to an increase in the elastic force significantly decreases the rate of macular growth during first few days of postnatal life (P1-P2). However, cytoplasmic Yap protein persists in supporting cells after cell-cycle exit (Figure 3; Figure 5). We believe that this fact explains why we do not observe a significant decrease in the total Yap protein between E17.5 and P1. To demonstrate a decrease in nuclear Yap protein, we would have to perform subcellular fractionation and to compare the amount of Yap protein from the nuclear fraction. Because of the limited amount of the sensory epithelia tissue, however, this experiment is not technically feasible. Instead, to demonstrate the loss of nuclear Yap signaling at P2, we have employed a more sensitive technique – quantitative PCR – and have demonstrated that the expression of genes encoding the downstream targets of nuclear Yap is downregulated by that stage. These new data 11 have been incorporated into Figure 3 the data file with the identical name.

Because E-cadherin, α-catenin, and gelsolin are not directly regulated by Yap signaling, we do not expect the expression of cognate genes to change sharply at P2. In fact, we do observe a significant increase in the expression of GSN but not in that of Cdh1 and Cnna between E17.5 and P2. As proposed by the Corwin group, we think that - independently of Yap signaling - postnatal accumulation of these and other inhibitors impedes nuclear Yap translocation and cell-cycle re-entry in the mature sensory epithelium

*2) Figure 4: The authors show an increase of proliferation and of Yap protein during regeneration upon injury of the utricule. These are only correlative observations and no proof that the observed proliferation depends on Yap, nor that Yap plays a role in regeneration.a) The authors should show, using the same canonical Hippo target genes they have used in Figure 3 that Yap activity is indeed increased during regenerationb) The authors should perform this experiment while blocking Yap function using for example the blocking peptide to show that regeneration is then blocked*

We would like to address point b first. We have added the requested experiments and demonstrated that supporting cell-cycle reentry upon injury in the neonatal utricle is blocked by infection with adenovirus expressing YTIP-GFP as opposed to control virus expressing only GFP. This experiment also demonstrated that the effect of YTIP is cellautonomous, for the proliferation of only GFP^+^, infected cells was affected. These data have been incorporated to Figure 4 and is described in the text as follows (Results section):

“We tested the effect of GFP-YTIP overexpression in the injury assay. […] However, the number of GFP-positive cells within the proliferating supporting-cell population was reduced significantly in GFP-YTIP cultures as compared to GFP controls (Figure 4).”

As for point a, we believe that the proposed experiment is not feasible. As we demonstrate in Figure 4, only 30-40 Sox2^+^ supporting cells in P4 neonatal utricle translocate Yap into their nuclei and re-enter the cell cycle after injury. At this stage, the utricle comprises approximately 8000 supporting cells (Figure 7). Therefore, the number of cells that respond to injury represents approximately 0.5% of the total. The change in expression of downstream target genes in damaged *versus* undamaged utricles is therefore unlikely to be detected by qPCR. However, to demonstrate that nuclear Yap translocation results in the activation of genes downstream of Yap, we have performed qPCR on the utricles cultured in low-stiffness and high-stiffness gel. Supplementing the data provided in Figure 3, these results have now been included in Figure 5.

*3) Figure 5: Some controls and a more in-depth analysis are required to convincingly show that Yap activity is solely responsible for increase proliferation and a doubling of the utricule in a less stiff gel. Also here,a) The authors should show, using the same canonical Hippo target genes they have used in Figure 3 that Yap activity is indeed increased in a softer gel as compared to a stiff gelb) The authors should use an alternative method, next to the blocking peptide, at best a yap1 mutant, to show that utricule area and proliferation also do not increase in a soft gel.*

We would like to clarify here that we are *not* claiming that Yap activity is solely responsible for supporting-cell proliferation. Results over the years have demonstrated that Wnt, EGF, and other signaling molecules are necessary for cellular proliferation in the inner ear. We do suggest, however, that the loss of nuclear Yap protein owing to an increase in the mechanical force and cellular density is a limiting factor on cellular proliferation during development.

To address point a, we have demonstrated that the expression of target genes downstream of nuclear Yap is significantly upregulated in the utricles cultured in 40 Pa gels as compared to 640 Pa gels. These data have been incorporated into Figure 5 and described in the text as follows:

“Whereas cell proliferation was limited to the organ’s periphery in 640 Pa gels by 4 d in culture, the supporting cells throughout the utricular macula reentered the cell cycle in 40 Pa gels (Figure 2). Yap labeling at the same time revealed that the pattern of the protein nuclear translocation in both condition was consistent with the pattern of supporting-cell proliferation (Figure 5—figure supplement 1). The expression of genes encoding the downstream targets of nuclear Yap, Ankrd1, Ctgf, and Cyr61, was also upregulated significantly in 40 Pa gels as compared to 640 Pa gels (Figure 5).”

We believe that owing to time limitations, the use of Yap-knockout animals is beyond the scope of current work. Instead, to address point b, we have used a small-molecule inhibitor, verteporfin, as an alternative blocker of nuclear Yap activity (Liu-Chittenden et al., 2012; Brodowska et al., 2014). Although we saw an effect on supporting-cell proliferation in the previously established concentration range (5-10 μM), we also observed hair-cell loss presumably owing to the compound’s toxicity. We therefore consider these results to be inconclusive and have not incorporated them into the manuscript, but provide them for the reviewer’s record.

*4) Figure 5 text: The last paragraph of subsection “Yap is necessary for supporting-cell proliferation”*

*"Unexpectedly, the supporting-cell density in GFP-YTIP-infected utricles was significantly lower than that in GFP-infected control cultures (Figure 5). As a result, although the area doubled, the total number of supporting cells in GFP-infected cultures increased only 20% in comparison with GFP-YTIP cultures (Figure 5). " This is very unclear. "Lower" should be "higher". The fact that "the total number of supporting cells in GFP-infected cultures increased only 20% in comparison with GFP-YTIP cultures" is to my understanding the cause and not the consequence of the higher density.*

Thank you for pointing out this mistake. We have corrected “lower” to “higher”. Additionally, because the total number of supporting cells was estimated from the cell density and the area of the utricles, this value did not provide any additional information and we have therefore removed it from the text and figure.

*"These data suggest that Yap activity is also required for cell spreading. Additional experiments are necessary to validate this prediction and to uncover the underlying mechanism." I think this statement is fully in the scope of this paper and the authors should show this increased spreading by measuring the cell area and the distance between individual cells.*

Because supporting cells never loose contact with one another, the average cellular area and the distance between the cells can be derived directly from measurements of cellular density. We therefore believe that including these data will not provide any additional information. The statement in question was meant to indicate a potential connection between nuclear Yap and other signaling pathways – such as Rho and JNK – that had been shown to control cellular size and spreading in the inner ear. We have removed this statement from the Results and expanded the Discussion to include this idea

*5) Figure 5—figure supplement 1: The authors should clarify in which panel of B, one looks at an apical or at a basal view. More importantly, hair cells in the utricule in 40Pa gel appear much bigger than in a 640Pa gel. The authors do not mention this observation. Given that Yap is not expressed in hair cells, how do the authors explain this?*

Thank you for pointing this out. All the panels in B represent apical views of the utricular macula. The basal views were provided in Figure 5. To avoid confusion, we have rearranged Figure 5 and Figure 5—figure supplement 1 to include an apical and a basal view in the same figure.

Although we did not characterize any hair-cell effects, we can offer the following explanation. We believe that variation in the size of a hair cell's apical surface does not necessarily reflect the change in size of the cell's soma. In the normally developing utricle such variation is observed owing to changes in cellular shape during hair-cell maturation. We believe this effect can be observed in Figure 3, in which some hair cells appear much larger in the apical view (lower panels) but their somata remain of constant size, as seen in the sections (top panels). An increase in the apical surfaces of hair cells might alternative result from passive stretching owing to a decrease in supporting-cell density in 40 Pa gels.

*6) Figure 6: It is not clear to me how the cell density can increase so much between E15.5 and E17.5* in vivo *(Figure 6) if both the supporting cell number (Figure 6) and the utricule area (Figure 1) increase at about the same speed. In this case, I would expect the cell density not to change much.*

According to our measurements, and as demonstrated in Figure 1, the area of the utricle doubles between E15.5 and E17.5, from 0.058 to 0.11 mm^2^. As shown in Figure 7, the number of supporting cells increases more than fourfold over the same period, from 0.9 to 4.5 thousand cells. Accordingly, the supporting cell density more than doubles, as demonstrated in Figure 7.

*7) a) It is not clear whether the authors interpretation is that both the decrease in cell density (or increased spreading) and increase in proliferation are driven by increased Yap activity or whether the decrease in stiffness results in decrease in cell density (yap independently) and the decrease in cell density, in turn, drives yap-dependent proliferation. This should be clarified in a precisely justified manner by the authors.*

Thank you for raising this point. We have clarified our hypothesis in the Discussion. As you point out, our model predicts that as utricle grows in the elastic environment of the inner ear, an increase in the elastic force opposing its expansion and creates a gradient in cellular density. Our model predicts that this effect is not dependent on nuclear Yap function, but instead that the elevated cellular density at the center of the utricle triggers the localization of cytoplasmic Yap and the protein's degradation, resulting in exit from the cell cycle.

*b) The initial question the authors stated in the Abstract and in the Introduction is concerning the link between inner ear growth arrest and inability to regenerate hair cells. Although the authors show that support for cell proliferation can be enhanced by lowering the elastic mechanical forces of the surrounding tissue, they do not show the consequences on hair cell regeneration. It would be very interesting that the authors damage the hair cells of ex-vivo cultured utricules in both 40Pa and 640Pa gels to compare their capacity to regenerate hair cells.*

We would like to clarify the question that we are trying to address in the manuscript. Our overall goal is to determine the mechanism that controls the rate and pattern of supporting-cell proliferation during utricular development. We further wish to determine the link between mechanical force and loss of Yap protein to arrest supporting-cell proliferation during development and cell-cycle re-entry during proliferative regeneration. We have rewritten parts of the Abstract, Introduction, and Discussion to make these points clearer. Although it would be very interesting to examine the regeneration of sensory receptors, to our knowledge *proliferative* hair-cell regeneration is not typically observed in the murine utricle in vitro. The supporting cells in the utricle of a young adult mouse retain a limited capacity to re-enter the cell cycle (Kawamoto et al., 2009; Golub et al., 2012; Burns et al., 2012a; Wang et al., 2015), but after mitosis these cells do not give rise to hair cells in vitro. EdU pulse-chase experiments demonstrate that the new sensory receptors that form in utricular cultures after hair-cell damage always originate from nonproliferative supporting cells. Therefore, the suggested experiment must be executed in vivo, where hair cells do form from dividing supporting cells. However, manipulation of tissue stiffness in such conditions would not be trivial. We do plan in our future experiments to assess hair-cell regeneration in vivowhile enhancing nuclear Yap function.

*Reviewer #3:*

*[…] Abstract*

*Please take into consideration that several labs (Forge, Raphael, Stone labs) have shown there is some spontaneous replacement of vestibular hair cells in adult rodents, probably due to supporting cell to hair cell conversion, and perhaps even in humans (Taylor et al., 2015). Plus, Warchol et al., (1993) showed that cultured adult utricles from guinea pigs and humans have considerable cell division in the macula (presumably of supporting cells) and Rubel et al., (1995) showed cell division leading to supporting cell replacement occurs in adult rodent vestibular epithelia.*

Although we agree with the reviewer on this point and are aware of the cited results, the first sentence of the Abstract is meant to be general. We believe that words “often” and “limited” indicate that there are instances in which hair-cell regeneration occurs in mammals. We do cite the work demonstrating hair-cell regeneration in the vestibular sensory organs of mammals in the Introduction.

*Add "the" in between "of" and "utricle's development".*

Thank you, we have done so.

*Introduction*

*The statement that hair cells almost never contact one another is misleading, especially since a recent paper (Pujol et al., 2014) showed that in the adult mouse utricle, a large proportion of the type II hair cell population is in contact via basolateral processes.*

This statement is meant to reflect the fact that, owing to lateral inhibition through NotchDelta signaling, two or more hair cells rarely arise directly adjacent to one another. Because the main point of the paragraph is to indicate the importance of supporting cells, however, we have rewritten the statement as follows:

“Hair cells are intercalated with supporting cells that are necessary for the proper sensory functions (Haddon et al., 1999)."

*In the fourth sentence the statements are untrue or misleading and should be modified. They suggest that supporting cell division does not proceed into the postnatal period and vestibular hair cells are not replaced after birth. However, Ruben (1967) showed that significant numbers of new vestibular epithelial cells are made postnatally; this paper should be cited. Further, Burns et al., (2012) showed continued macular cell proliferation into the neonatal period and production of several hair cells from the products of those dividing cells. The wording suggests that the authors have not read several papers showing that there is considerable regeneration of vestibular hair cells in adult rodents, probably by direct transdifferentiation. By these accounts, percentages of regenerated hair cells range from 20% to 70% of normal levels (in some regions). Several references should be cited, including Forge et al., 1993; Forge et al., 1998; Kawamoto et al., 2009; Lin et al., 2011; Golub et al., 2012; and Slowik et al., 2013. Further, these findings should be taken into consideration and addressed in the Discussion in the context of this paper's findings during development.*

We wish to clarify our point. We are aware of the pioneering work of Ruben and more recent studies on the addition of new hair cells in the vestibular sensory organs in mammals. We also appreciate the literature on hair-cell regeneration in adult rodents. However, all of the existing work on the topic, including the citations suggested by the reviewer, demonstrates that new hair cells form only through *transdifferentiation of supporting cells*. Although we agree that it is important to acknowledge this work, the whole focus of the Introduction and indeed of the entire manuscript is on the proliferation of supporting cells. Because supporting cells fail to re-enter cell cycle after hair cells have been lost in adult sensory epithelia of mammals, as demonstrated in most of the suggested citations, we stand by our statement that mammals are the only vertebrates that cannot *fully* recover from hearing and balance deficits.

On the reviewer’s comment about the cited work of Corwin’s group, although hair cells are actively added in the murine utricle up until the end of the first week after birth – and at slow rates thereafter – as demonstrated by Burns et al., in 2012, supporting cell *proliferation* ceases around P1-P2. We have conformed these observations in our past (Gnedeva et al., 2015) and current work.

Overall, we agree with the reviewer’s point and have rewritten the first paragraph of the Introduction to include the suggested literature on hair-cell regeneration through direct conversion of supporting cells. The paragraph now reads:

“Additionally, when hair cells are lost in nonmammalian species, supporting cells can proliferate and transdifferentiate into new sensory receptors, allowing for recovery of hearing and balance (Corwin and Cotanche, 1988; Ryals and Rubel, 1988; Harris et al., 2003; Taylor and Forge, 2005). Although supporting cells in the vestibular sensory organs of adult mammals retain a limited ability to regenerate hair cells through direct conversion (Ruben, 1967; Forge et al., 1993; Rubel et al., 1995; Kawamoto et al., 2009; Lin et al., 2011; Golub et al., 2012), they lose the ability to reenter the cell cycle after neonatal stages (Golub et al., 2012; Burns et al., 2012a; Wang et al., 2015), making hearing and balance loss irreversible in mammals.”

We also realized that when we talk about regeneration later in the manuscript, we have to clarify that we mean proliferative regeneration in particular. We have corrected our statements throughout the manuscript to avoid confusion.

*Second paragraph, third sentence. Since White et al., (2012) saw the EGF effect in chicken auditory organs, I do not believe this sentence helps. Several groups have examined EGF signaling in adult mouse utricles (e.g., Oesterle's group and Gao's group). Although White et al., assessed effects of EGF on mouse cochlear supporting cells, they were dissociated cells, not intact epithelia.*

We did refer to the EGF effects on dissociated supporting cells of the organ of Corti. The same effect was shown earlier by Doetzlhofer et al., in 2004. To the best of our knowledge, Oesterle's group and Gao's group demonstrated the effect of heregulin, rather than EGF, on supporting-cell proliferation. To include this work, we have broadened our statement as follows:

“ErbB signaling is also involved, for EGF and heregulin enhance supporting-cell proliferation in vitro and inhibition of the EGFR pathway arrests mitotic activity in sensory epithelia (Zheng et al., 1999; Hume et al., 2003; Doetzlhofer et al., 2004; White et al., 2012).”

*I would consider mentioning here work from the Pirvola and the Burns groups on the failure of cell cycle effectors such as cyclin D to promote division in mouse utricles. This seems quite germane to this discussion.*

We would like to thank the reviewer for bringing up this point. Additionally, although P27Kip1-knockout animals are characterized by prolonged proliferation in the sensory organs of the inner ear, the prosensory cells do exit the cell cycle in these animals. These data, in conjunction with works of the Pirvola’s laboratory, suggest that upregulation of inhibitors of cyclin-dependent kinase is not the sole mechanism by which growth and cell-cycle re-entry during regeneration are suppressed in mammals. Inasmuch as these results reinforce our argument, we have added the appropriate text and citations.

*Results section. Why assume supporting cells secrete a growth-inhibiting molecule? Why not hair cells, or why couldn't supporting cells secrete a growth-promoting molecule? Would the observations still rule out a morphogen model if the morphogen were growth-promoting?*

If hair cells were to secrete the growth-inhibiting molecule our results would not be modified substantially. Because hair cells are intercalated with supporting cells, the resulting concentration gradient of the morphogen would be almost identical, and the conclusions drawn from our current model would follow. In particular, for a utricle in a low-stiffness gel the concentration of the morphogen at the center of the macula would still exceed the threshold, and therefore we would not observe any cellular proliferation in this region.

A growth-promoting molecule would not be compatible with our experimental observations. The shape of the gradient would remain unchanged whether the molecule in question were growth-inhibiting or growth-promoting, with a concentration highest at the center of the utricle and progressively lower towards the periphery (Figure 1). This pattern would mean that proliferation would be promoted at a higher rate at the center of the utricle and little proliferation would be observed at the boundary. Such a result would directly contradict our EdU experiments, in which we observe that proliferation occurs primarily at the periphery of the utricle (Figure 7). Furthermore, such a model would lead to an ever-expanding utricle, unless for some additional mechanism arrests the secretion of the growth-promoting molecule.

*Figure 7 disagree on several points here; what is stated does not match the figure. First, cytoplasmic Yap labeling looks strong in the images. I think it is clear from the images in Figure 3 that no nuclear Yap is seen in the central region while some nuclear Yap is seen in the periphery. It is clear however, that some cytoplasmic Yap is retained in the periphery, which makes it a bit hard to discern.*

We agree with the reviewer. We have removed “weak” from the description of Yap labeling at the center of the utricle. We have also changed the wording to reflect that some nuclear Yap is seen in the periphery. Similar to the Notch intracellular domain, which we know can be translocated into nuclei but cannot be visualized there immunohistochemically, it is rather hard to clearly observe nuclear Yap in vivo.

*Subsection “Yap signaling during development and regeneration of the murine utr”. Need to add Kawamoto et al., 2009 to Golub et al., 2012 to reference list.*

Thank you, we have added the citations for Kawamoto et al., 2009; Burns et al., 2012; and Wang et al., 2015.

*Subsection “Yap signaling during development and regeneration of the murine utr”, Figure 4. This is an odd experiment to perform, as opposed to targeted hair cell ablation, which could be accomplished with neomycin* in vitro *using published methods. One concern is that the authors are not actually examining cell divisions related to sensory epithelial cell repair. Were the EdU^+^ cells Sox2^+^? This would indicate if the cells dividing were indeed supporting cells or some other cell, such as from the stroma or the transitional epithelium. Did the authors investigate the fate of the post-M cells in that region? This would indicate divisions are leading to hair cells. I see this as a major problem with interpreting the data.*

After neomycin treatment, supporting cell-cycle reentry is poorly synchronized, as the onset of mitotic activity spreads for days after hair cells have been lost. In view of the dynamic shuttling of Yap protein between the cytoplasm and nucleus, it is therefore harder to demonstrate the concurrence of nuclear Yap localization and supporting-cell proliferation after aminoglycoside assault. For the purposes of our current work, we focus on the ability of supporting cells to re-enter the cell cycle, rather than on hair-cell regeneration. Because the ability of supporting cells to change shape and undergo mitosis after mechanical injury has been well correlated with the proliferative response to hair-cell loss by the Corwin group, we believe this assay to be an appropriate means of rapidly evoking supporting-cell proliferation. We have added high-magnification images to demonstrate clearly that the cells that translocate Yap into their nuclei and enter the cell cycle in our injury assay are Sox2^+^ (Figure 4).

Although it would be very interesting to examine the regeneration of the sensory receptors, to our knowledge, *proliferative* hair-cell regeneration is not typically observed in vitro. EdU pulse-chase experiments by Burns et al., 2012a and Wang et al., 2015 demonstrate that the new sensory receptors formed in utricular cultures after hair-cell damage are not EdU^+^. The suggested experiment must therefore be executed in vivo, which owing to time limitations is beyond the scope of the current work.

*Subsection “Yap is necessary for supporting-cell proliferation”. No – the density was higher! Did you mean macular area? Overall, this section is confusing. Here is what the data in Figure 5 say to me: In YTIP-treated epithelia, macular area was halved (E) and dividing cell numbers were reduced to ~20% of normal (H) while supporting cell density doubled (G). This is not surprising to me, but I don't understand why this suggests that cell spreading was reduced. Further, it is strange to me is that epithelial cell numbers were not dramatically reduced, given the EdU data.*

We apologize for the poor wording: this paragraph was unclear to other reviewers as well. We have rewritten the text to reflect the data in the figure. When we discuss areal measurements, we always refer to the areas of sensory epithelia. We have clarified this point throughout the text. As for change in the cellular size and spreading, we originally found it surprising that cellular density is affected in YTIP virus-infected cultures. Our model and data, as well as the existing literature on Hippo signaling, suggest that cellular density regulates subcellular Yap localization and not *vice versa*. That is why the fact that cell density is higher in the YTIP condition surprised us. It is possible, though, that the loss of cellular proliferation changes the mechanical properties of the sensory epithelium or that nuclear Yap also regulates cellular size and spreading as was shown in Tumaneng et al., 2012. We have rewritten the text to clearly reflect the observed results.

*Subsection “Mechanical force confines cellular proliferation to the utricle’s periphery”. Utricular growth does not cease at P2. Burns et al., 2012 (Figure 1) show macular area increases significantly out to P8.*

Burns et al., 2012 demonstrated and we later confirmed (Gnedeva et al., 2015) that supporting-cell proliferation ceases by P2. The discrepancies in the dynamics of the areal growth between the work of Burns et al., 2012 and our manuscript arise from the ways in which macular areas are determined. Because we are interested in the dynamics of supporting-cell proliferation, we measure the utricle’s area as the area occupied by Sox2^+^ cells. When evaluating the dynamics of hair-cell formation, Burns et al., measured the macular area occupied by hair cells. Because many new hair cells arise at the utricle’s periphery between P2 and P8, the area of the organ appears by that measure to grow significantly over the same period.

*Discussion section This is a short Discussion.*

We have expanded the Discussion to include some of the suggestions. Please see below our detailed responses to the particular points.

*The authors should consider addressing the following as well:*

*Does Yap play a similar role in limiting growth of any other sensory or neural tissues (retina, olfactory epithelium, other)? It would be a good addition to discuss this here, briefly.*

In the Discussion, we focus on explaining the connection between elastic force and the arrest of tissue growth. Yap signaling was shown to regulate cellular proliferation during retinal development (Kim at al., 2016), but it was also found to control the proliferation of many other tissues. We find that focusing the Discussion on the mechanical control of the pattern of supporting-cell proliferation and the arrest of utricular growth helps us to outline the novelty of our work and the approaches used. We have expanded the Discussion to include other known examples of mechanical control of growth and their connection to Yap-Hippo signaling.

*Is the favoured model consistent with all forces controlling cell division being restricted to the sensory epithelium (i.e., related to epithelial cell density)? Might some forces be derived from outside the macula? If so, what could they be?*

The forces constraining growth arise both from the transitional epithelium surrounding the sensory epithelium and the surrounding mesenchymal stroma and cartilage. We acknowledge that this point was not clear in our previous version of the manuscript; we have expanded the explanation of our model and added Figure 1—figure supplement 1 to make this point clear.

*What significance is the boundary of transitional epithelial (TE) cells with respect to limiting supporting cell division and growth?*

As for the previous question, in our model the transitional epithelium is part of the tissue that limits growth and division. We have updated the text to better explain this point.

*What is Yap expression like in the TE?*

Because we did not label the utricles with markers for the transitional epithelium, we cannot comment on this point with confidence. Based on the position and lack of Sox2 labeling, however, we can attest that, when transitional epithelium cells actively proliferate in culture, we see a high level of Yap expression (Figure 4). Yap expression is not specific to the sensory epithelium; in fact, in the developing embryo Yap is expressed in most tissues. We therefore do not see how this point might benefit the Discussion.

*Why might supporting cell division eventually cease? What changes in the stiffness of cell density might allow this to occur?*

Thanks to this suggestion, we have included this point in the Discussion. In order to divide, cells must first increase in size. In our model the elastic force generated by the surrounding tissue physically restricts cellular growth and results in increased cellular density Our current understanding of the system indicates that, after this physical restriction has slowed growth and cellular division, loss of nuclear Yap signaling completely arrests cellular proliferation.

*Do you feel you have ruled out the possibility that both types of forces – morphogen and elasticity – might be actively controlling utricular size?*

Our elasticity-limited and morphogen-limited models are non-exclusive and in principle a combination of the two effects could be present. Given the striking growth in macular area and the extensive number of cell divisions that occur at the center of utricles in the low-stiffness collagen experiments, we believe that any growth-inhibiting molecule plays only a minor role in setting the final size of the utricular sensory epithelium.

*During damage, how might extrusion or degradation of hair cells alter Hippo signaling?*

This is very interesting point. In fact, we think that in nonmammalian species, the signal that induces supporting cells to re-enter the cell cycle after hair-cell loss might be mechanical in nature. It would be interesting to examine the expression of Yap in the regenerating avian basilar papilla in addition to the mammalian utricle. We have included this discussion in the text.

*Video 3: This seems like new data; why isn't it included in the Results section?*

Because LGR5^+^ cells are believed to be the stem cell-like population in the developing inner ear, we found it important to demonstrate a model that incorporates stem cells. The model invoking stem cells as the only source of proliferation results in a pattern of cell divisions inconsistent with any of our experimental observations. We found it difficult to find an appropriate spot for these results in the manuscript. As suggested, we have moved a description of the stem-cell model to the Results section entitled “The effect of elastic force on the pattern of supporting-cell proliferation.

*Methods*

*No methods were provided for the following: Western blots, RNAseq, virus experiments! For the Western and RNAseq, were sensory epithelia only used or were whole utricles used?*

Thank you for pointing this out. We have added descriptions of the experimental procedures for Western blotting, qPCR (now used in Figure 3 and Figure 5), and viral experiments to the Methods. In the current work, we use some of the RNA-sequencing data acquired in Gnedeva et al., 2015. This publication is cited in the text. We have added a sentence commenting on this to the Methods.

Subsection “Mechanical force confines cellular proliferation to the utricle’s periphery”:

*- What% of the macula was sampled?*

*- Were cells from each zone (striola, lateral extrastriola, medial extrastriola) sampled?*

*- In utricles, type II hair cells are also Sox2^+^; how did you assure you were not counting Sox2^+^ hair cells? (This is particularly an issue with manual counting methods).*

*- What was the degree of error with automated areal measurements? (This should be known, since the algorithms were compared to manual counts).*

For the counts provided for utriclular development, we did not sample different areas of the utricular macula. Instead, as described in the Methods, we created a highresolution image of whole utricles, and counted *all* the Sox2^+^ supporting cells. Sox2^+^ hair cells were excluded on the basis of Myo7A labeling. We have included these details in the Methods. By manual counting the cells at one of the developmental times, we found the errors of our segmentation algorithm to be consistently in the range of 3-5%.

*Subsection “Mechanical force confines cellular proliferation to the utricle’s periphery”. There is no Figure 4. What does this mean: "calculated the mean outline?" Is this area?*

*Subsection “Mechanical force confines cellular proliferation to the utricle’s periphery”. There is no Figure 4. Since I cannot link this information to any figure, I cannot interpret or assess this paragraph.*

We have corrected this error in the text and clarified the procedure. The illustration formerly referred as Figure 4 is now Figure 7 and Figure 4 is now Figure 7.

*Figure 2, here, and for several places throughout the manuscript: N's should be provided for each group (e.g., N=3 for 40 Pa and N=3 for 640 Pa)*

The same number of replicates is used for each experimental group as a rule. The instances in which the number of replicates varies between groups are indicated in the figure legends. We have corrected this throughout the text.

*– Figure 2 source data file is very confusing. I opened the files in Excel. First, it states supporting cell area instead of epithelial area. Second, it is not clear which group (40, PA 640 PA or* in vivo*) is described in each column, since there seem to be column titles missing for each column. Column B has no heading at all.*

We apologize for these errors. We have corrected all the data files and ensured that the data are properly labeled.

*–Figure 2 source data. Again, I cannot see the column titles, so I am not sure they are there. I recommend carefully checking all source data to make sure they are complete (have all headings) and match the terms used in the figures and text.*

We apologize for these errors. We have corrected all the data files and ensured that the data are properly labeled.

*Figure 3, are you confident the cells with the higher Yap intensity are in the sensory epithelium, or might they be transitional epithelial cells? What are the units for fluorescence intensity?*

*It is impossible to tell that hair cells do not express Yap. The reader would be more convinced if fluorescence were shown to not overlap with cytoplasmic hair cell marker labelling. Another possibility is to reference Figure 5—figure supplement 1, since it is quite clear there that hair cell cytoplasm isn't labelled.*

The pattern of supporting-cell proliferation is very stable in the utricle; in Figure 7 one can see that the band of EdU^+^ cells at the organ’s periphery is positive for Sox2. Although we were not able to achieve quadruple labeling for Yap, GFP, EdU, and Sox2, judging by the epithelial thickness and the immediate proximity to hair cells, we are certain that the proliferating cells with a higher Yap intensity at the periphery of the utricle are supporting cells.

The intensity was determined using ImageJ application where it is expressed as a relative unitless measure. In a single channel image, the intensity of a black pixel is 0 (Min) and the intensity of a saturated pixel is 255 (Max). For the RGB images, used in this case, maximum intensity of a pixel in each channel is 85 (255/3). We have added this information to the Methods section and to the figure legends, where appropriate.

*Figure 4 legend, regarding the statement "The same cells enter the cell cycle". This is true for some cells, but lots of dividing cells seem NOT to have translocated nuclear Yap. This could be misleading or suggest a bias. These images should be shown at higher magnification, to better enable readers to see the Yap and EdU labelling in the nuclei.*

We have added high-magnification images to ensure that readers can judge for themselves whether Yap is translocated to the nuclei of a subset of Sox2^+^ supporting cells at the injury site (Figure 4, lower panels). We agree with the reviewer that not all of the proliferating supporting cells demonstrate nuclear Yap labeling. However, because the tissue was allowed to recover in medium containing EdU for a prolonged period of time, the supporting cells are likely to be in different phases of the cell cycle. Because Yap shuttling between cell nucleus and cytoplasm is dynamic, this temporal dispersion may explain the differences in the subcellular Yap localization. To directly demonstrate the role of nuclear Yap in supporting-cell proliferation after injury, we have conducted the same experiment while blocking nuclear Yap function with YTIP peptide. The results are included into Figure 4 and clearly demonstrate the decrease in number of Sox2/EdU/GFP triple-positive cells in GFP-YTIP condition as compared to GFP controls.

*Figure 5, did the authors examine the cell-autonomous nature of the effect of GFP-YTIP? Do they know if there was a significant change in the GFP+ versus GFP- cells? This would be very important to understand, as it could expand our understanding of how Yap functions.*

*– I think it's important to note that the supporting cell density decreased with low stiffness (Figure 5). How about effects on total supporting cell numbers?*

It is hard to determine whether the effect of Yap inhibition is cell-autonomous in organotypic bubble cultures. Because the system is closed, the loss of cellular proliferation in a subset of infected cells likely changes the overall mechanical properties of the tissue. In other words, loss of nuclear Yap and cellular proliferation in a subset of cells increases the elastic force on noninfected cells.

As now shown in Figure 4, we have added an experiment examining the ability of supporting cells to re-enter the cell cycle after injury while nuclear Yap was blocked. Because the cells are not restricted by an elastic boundary at the injury site, this experiment allowed us to determine whether the nuclear Yap effect is cell-autonomous. By analyzing the GFP^+^ and GFP^-^ populations of Sox2^+^ supporting cells, we demonstrated that the effect of YTIP on cell proliferation can be seen only in GFP^+^ cells, suggesting that the effect of nuclear Yap is cell-autonomous.

As suggested, we have noted the supporting-cell density in the description of panel C. Owing to the technical difficulty in dissecting an intact utricle from a collagen gel after three days in culture, we could not count every supporting cell in the large number of cultured utricles to provide statistically relevant numbers as we did for the developmental stages. We therefore can only estimate the overall number of supporting cells from the supporting-cell density and the utricular area. Because these data seem redundant, we chose not to include them in the figures.

*Figure 6 do not understand what is being shown here. What is the dotted line? I do not see any shaded area around the line. What does x axis represent?*

*– Figure 6, how were the center and the periphery defined for these counts? How was this unbiased? Are data in Figure 6, and I from real utricles or simulated data?*

*– The authors should clearly state which cells were counted by including "supporting" or "macular" in the y-axis labels whenever possible.*

*– What is significance of asterisks in the graphs? This should be explained in the legend.*

The figure legend and the description in the Results section associated with Figure 6 (now Figure 7) have been clarified and rewritten. The results of the EdU-labeling experiment shown in panel F (now E), were added to demonstrate the consistency of the pattern of supporting-cell proliferation. The images for EdU and Sox2 immunolabeling for five E18.5 utricles were overlayed to determine the average macular boundary and the average position of proliferating cells. One can see in panel D a 75µm band at the macular periphery, within which supporting cells proliferate at E18.5. The graphic representation of the same data is shown in panel E. The abscissa demonstrates the distance from the macular boundary, with negative values representing the direction towards the macular center. The vertical dashed line accordingly represents the macular boundary. The peak of the distribution of EdU labeling, which represents the position at which most of the proliferating supporting cells are located, lies immediately inward from the macular boundary.

Based on these data, in panels G and I (now panel C), we defined the periphery as the 50µm band of Sox2^+^ cells nearest the macular border. The border of the central region was defined as being at least 100µm from any of the macular edges. All the counts are of the real utricles in which the number of Sox2^+^ cells was determined in 50µm x 50µm squares in the peripheral or central regions and are expressed as cellular densities (cells per mm^2^). We have added these descriptions to the manuscript. We have followed the standard nomenclature for statistical significance: one asterisk represents p < 0.05, two represent p < 0.01, and three denote p < 0.001. Although “supporting” is not included in some of the y-axes labels to avoid clattering, this information is clearly indicated in the figure legend.

[Editors' note: further revisions were requested prior to acceptance, as described below.]

*The manuscript has been substantially improved and reviewers 1 and 2 are happy that their queries and concerns have been addressed. Reviewer 3 is very positive overall but has some remaining suggestions concerning wording or comparison to published studies. We are therefore returning the manuscript to give you the option of addressing or rebutting these concerns, which should be quick to do. The detailed comments are included below.*

*Reviewer #3:*

*This excellent paper has been significantly enhanced by the added experiments and the careful editing of the text. I have only a few additional suggestions before publication.*

We would like to thank the reviewer again for such a remarkably detailed analysis of our original manuscript and the resubmission. We appreciate the time and effort reviewer had spent on the analysis of our paper and we are grateful for reviewer’s high opinion of our work.

*Abstract. I recommend rephrasing the first sentence to "limited ability of cells in the inner ear to proliferate", as inner ear proliferation would suggest duplications of the organ.*

Thank you, we have incorporated the suggested change.

*Introduction. I am in favor of the wording now, with two small gripes.*

*The Warchol et al., 1993 Science paper refutes the last statement that "supporting cells [...] lose the ability to reenter the cell cycle after neonatal stages", since they showed that cultured utricles from adult humans and guinea pigs attained numerous tritiated thymidine-labeled supporting cells over time. I realize this is a stand-out paper, but I wonder if you could add to the reference list, "but see Warchol et al., 1993". This applies to text in other places in the manuscript (e.g., subsection “Yap signaling during development and regeneration of the murine utricle”).*

We have added the suggested citation in the Introduction. Because on subsection “Yap signaling during development and regeneration of the murine utricle” we describe the development of the mouse utricle in particular, this citation is irrelevant there. We have added “murine” to the sentence to make our point clearer.

*I recommend switching the last sentence, which reads "making hearing and balance loss" to "and may be a primary reason why hearing and balance function fail to recover in mammals after hair cell damage". In my opinion, it is too bold to suggest that the lack of proliferation is THE cause for the lack of functional recovery.*

We agree with the reviewer and have incorporated the suggested change.

*Subsection” Yap signaling during development and regeneration of the murine utricle”. Please remove Kawamoto et al., and Golub et al., from this list; only experiments on adult mice were performed in these studies.*

*Subsection “The effect of elastic force on the pattern of supporting-cell proliferation”. I originally disagreed with the statement that utricular growth ceases at P2, and I still find information confusing, since no changes were made to the text.*

We agree with this comment and have removed the suggested citations.

*It's important to distinguish between cell division and organ growth. Although cell division may cease at P2, the organ could continue to grow beyond that point. Indeed, Burns et al., 2012b found evidence to support that the utricle does grow in size and hair cell numbers increase after P2. I recommend removing the reference to Burns et al., 2012b here, or providing a clearer explanation of the disparate conclusions in the two papers, as you did in your response to reviewers.*

*I will also note that Figure 1 of Burns et al., 2012a and Figure 4 of Burns et al., 2012b show BrdU incorporation in utricular macula cells after P2. So, I am also confused by the declarations throughout that supporting cell proliferation ceases within the first two days of postnatal life.*

*These may seem like petty gripes, but it's important to be accurate, and I feel some findings are stated as absolutes that are misleading.*

We see the reviewer’s point. We chose not to remove the citation to the work of Burns et al., 2012, as it is a pioneering study on the dynamics of utricular growth and in our opinion must be cited in subsection “The effect of elastic force on the pattern of supporting-cell proliferation”. We have instead corrected our statement that the “utricle ceases to grow by P2” to “The growth of the murine utricle and supporting-cell proliferation decline dramatically within the first two days of postnatal life” in subsection “Yap signaling during development and regeneration of the murine utricle “to “supporting-cell proliferation decreases dramatically by that stage” in subsection “The effect of elastic force on the pattern of supporting-cell proliferation”, and to “supporting-cell proliferation ceases” in the Discussion section. We agree that growth and proliferation are related but distinct processes and that the statement needed be clarified throughout the text.